# NEO: **Non Equilibrium Sampling on the Orbit of a Deterministic Transform**

**Achille Thin**
Centre de Mathématiques Appliquées
École polytechnique
Palaiseau, France
achille.thin@polytechnique.edu

**Yazid Janati**
Samovar
Télécom SudParis, Département CITI
Palaiseau, France
yazid.janati_elidrissi@telecom-sudparis.eu

**Sylvain Le Corff**
Samovar
Télécom SudParis, Département CITI
Palaiseau, France

**Charles Ollion**
Centre de Mathématiques Appliquées
École polytechnique, Palaiseau, France

**Arnaud Doucet**
Department of Statistics,
University of Oxford, UK

**Alain Durmus**
Université Paris-Saclay, ENS Paris-Saclay, CNRS,
Centre Borelli, F-91190 Gif-sur-Yvette, France

**Eric Moulines**
Centre de Mathématiques Appliquées
École polytechnique, Palaiseau, France

**Christian Robert**
Ceremade, Université Paris-Dauphine, France
& Department of Statistics, Warwick University, UK

## Abstract

Sampling from a complex distribution $\pi$ and approximating its intractable normalizing constant Z are challenging problems. In this paper, a novel family of importance samplers (IS) and Markov chain Monte Carlo (MCMC) samplers is derived. Given an invertible map T, these schemes combine (with weights) elements from the forward and backward Orbits through points sampled from a proposal distribution $\rho$. The map T does not leave the target $\pi$ invariant, hence the name NEO, standing for Non-Equilibrium Orbits. NEO-IS provides unbiased estimators of the normalizing constant and self-normalized IS estimators of expectations under $\pi$ while NEO-MCMC combines multiple NEO-IS estimates of the normalizing constant and an iterated sampling-importance resampling mechanism to sample from $\pi$. For T chosen as a discrete-time integrator of a conformal Hamiltonian system, NEO-IS achieves state-of-the art performance on difficult benchmarks and NEO-MCMC is able to explore highly multimodal targets. Additionally, we provide detailed theoretical results for both methods. In particular, we show that NEO-MCMC is uniformly geometrically ergodic and establish explicit mixing time estimates under mild conditions.

35th Conference on Neural Information Processing Systems (NeurIPS 2021).

# 1 Introduction

Consider a target distribution of the form $\pi(x) \propto \rho(x)\mathrm{L}(x)$ where $\rho$ is a probability density function (pdf) on $\mathbb{R}^d$ and $\mathrm{L}$ is a nonnegative function. Typically, in a Bayesian setting, $\pi$ is a posterior distribution associated with a prior distribution $\rho$ and a likelihood function $\mathrm{L}$. Another situation of interest is generative modeling where $\pi$ is the distribution implicitly defined by a Generative Adversarial Networks (GAN) discriminator-generator pair where $\rho$ is the distribution of the generator and $\mathrm{L}$ is derived from the discriminator [31, 6]. In a Variational Auto Encoder (VAE) context [14, 5], $\pi$ could be the true posterior distribution, $\rho$ the approximate posterior distribution output by the encoder and $\mathrm{L}$ an importance weight between the true posterior and approximate posterior distributions. We are interested in this paper in sampling from $\pi$ and approximating its intractable normalizing constant $\mathrm{Z} = \int \rho(x)\mathrm{L}(x)\mathrm{d}x$. These problems arise in many applications in statistics, molecular dynamics or machine learning, and remain challenging.

Many approaches to compute normalizing constants are based on Importance Sampling (IS) - see [1, 2] and the references therein - and its variations, among others, Annealed Importance Sampling (AIS) [19, 34, 10] and Sequential Monte Carlo (SMC) [9]. More recently, Neural IS has also become very popular in machine learning; see e.g. [11, 18, 21, 22, 32, 33]. Neural IS is an adaptive IS which relies on an importance function obtained by applying a normalizing flow to a reference distribution. The parameters of this normalizing flow are chosen by minimizing a divergence between the proposal and the target (such as the Kullback–Leibler [18] or the $\chi^2$-divergence [1]). Recent work on the subject proposes to add stochastic moves in order to enhance the performance of the normalizing flows [33].

More recently, the *Non-Equilibrium IS* (NEIS) method has been introduced by [23] as an alternative to these approaches. Similar to Neural IS, NEIS consists in transporting samples $\{X^i\}_{i=1}^N$ from a reference distribution using a family of deterministic mappings. For NEIS, this family is chosen to be an homogeneous differential flow $(\phi_t)_{t \in \mathbb{R}}$. In contrast to Neural IS, for any $i \in [N]$, the sample $X^i$ is propagated both forward and backward in time along the orbits associated with $(\phi_t)_{t \in \mathbb{R}}$ until stopping conditions are met. Moreover, the resulting estimator of the normalizing constant is obtained by computing weighted averages of the whole orbit $(\phi_t(X^i))_{t \in [\tau_{+,i}, \tau_{-,i}]}$, where $\tau_{+,i}, \tau_{-,i}$ are the resulting stopping times, and not only the endpoints $\phi_{\tau_{+,i}}(X^i), \phi_{\tau_{-,i}}(X^i)$. In [23], the authors provide an application of NEIS with $(\phi_t)_{t \in \mathbb{R}}$ associated with a conformal Hamiltonian dynamics, and reports impressive numerical results on difficult normalizing constants estimation problems, in particular for high-dimensional multimodal distributions.

We propose in this work NEO-IS which alleviates the shortcomings of NEIS. Similar to NEIS, samples are drawn from a reference distribution, typically set to $\rho$, and are propagated under the forward and backward orbits of a *discrete-time* dynamical system associated with an invertible transform $\mathrm{T}$. An estimator of the normalizing constant is obtained by reweighting all the points on the whole orbits using the IS rule. Contrary to NEIS, the NEO-IS estimator of $\mathrm{Z}$ is unbiased under assumptions that are mild and easy to verify. It is more flexible than NEIS because it does not rely on the accuracy of the discretization of a continuous-time dynamical system.

We then show how it is possible to leverage the unbiased estimator of $\mathrm{Z}$ defined by NEO-IS to obtain NEO-MCMC, a novel massively parallel MCMC algorithm to sample from $\pi$. In a nutshell, NEO-MCMC relies on parallel walkers which each estimates the normalizing constant but are allowed to interact through a resampling mechanism.

Our contributions can be summarized as follows.

(i) We present a novel class of IS estimators of the normalizing constant $\mathrm{Z}$ referred to as NEO-IS. More broadly, a small modification of this algorithm also allows us to estimate integrals with respect to $\pi$. Both finite sample and asymptotic guarantees are provided for these two methodologies.

(ii) We develop a new massively parallel MCMC method, NEO-MCMC. NEO-MCMC combines NEO-IS unbiased estimator of the normalizing constant with iterated sampling-importance resampling methods. We prove that it is $\pi$-reversible and ergodic under very general conditions. We derive also conditions which imply that NEO-MCMC is uniformly geometrically ergodic (with an explicit expression of the mixing time).

**(iii)** We illustrate our findings using numerical benchmarks which show that both NEO-IS and NEO-MCMC outperform state-of-the-art (SOTA) methods in difficult settings.

## 2 NEO-IS algorithm

In this section, we derive the NEO-IS algorithm. The two key ingredients for this algorithm are (1) the reference distribution $\rho$ and (2) a transformation T assumed to be a $C^1$-diffeomorphism with inverse $T^{-1}$. Write, for $k \in \mathbb{N}^* = \mathbb{N} \setminus \{0\}$, $T^k = T \circ T^{k-1}$, $T^0 = \mathrm{Id}_d$ and similarly $T^{-k} = T^{-1} \circ T^{-(k-1)}$. For any $k \in \mathbb{Z}$, denote by $\rho_k : \mathbb{R}^d \to \mathbb{R}_+$ the pushforward of $\rho$ by $T^k$, defined for $x \in \mathbb{R}^d$ by $\rho_k(x) = \rho(T^{-k}(x))\mathbf{J}_{T^{-k}}(x)$, where $\mathbf{J}_\Phi(x) \in \mathbb{R}^+$ is the absolute value of the Jacobian determinant of $\Phi : \mathbb{R}^d \to \mathbb{R}^d$ evaluated at $x$. In line with multiple importance sampling *à la* Owen and Zhou [20], we introduce the proposal density

$$\rho_T(x) = \Omega^{-1} \sum\nolimits_{k \in \mathbb{Z}} \varpi_k \rho_k(x) , \tag{1}$$

where $\{\varpi_k\}_{k \in \mathbb{Z}}$ is a nonnegative sequence and $\Omega = \sum_{k \in \mathbb{Z}} \varpi_k$. Note that we assume in the sequel that the support of the weight sequence defined as $\{k \in \mathbb{Z} : \varpi_k \neq 0\}$ is finite. Thus, the mixture distribution in (1) is a **finite mixture**. Given $x \in \mathbb{R}^d$, $\rho_T(x)$ is a function of the forward and backward orbit of T through $x$.

For any nonnegative function $f$, the definition of $\rho_T$ implies that $\int f(y)\rho_T(y)\mathrm{d}y = \Omega^{-1} \int \sum_{k \in \mathbb{Z}} \varpi_k f(T^k(x))\rho(x)\mathrm{d}x$. Assuming that $\varpi_0 > 0$, the ratio $\rho(x)/\rho_T(x) \leq \varpi_0^{-1}\Omega < \infty$ is bounded. We can therefore apply the IS principle which allows to write the identity

$$\boxed{\int f(x)\rho(x)\mathrm{d}x = \int \left( f(y)\frac{\rho(y)}{\rho_T(y)} \right) \rho_T(y)\mathrm{d}y = \int \sum_{k \in \mathbb{Z}} f(T^k(x))w_k(x)\rho(x)\mathrm{d}x ,} \tag{2}$$

where the weights are given by (see Appendix A.2 for a detailed derivation),

$$w_k(x) = \varpi_k \rho(T^k(x))/\{\Omega\rho_T(T^k(x))\} = \varpi_k \rho_{-k}(x) \bigg/ \sum\nolimits_{i \in \mathbb{Z}} \varpi_{k+i}\rho_i(x) . \tag{3}$$

We assume in the sequel that $\varpi_0 > 0$. In particular, note that under this condition, the weights $w_k$ are also upper bounded uniformly in $x$: for any $x \in \mathbb{R}^d$, $w_k(x) \leq \varpi_k/\varpi_0$. Equations (2) and (3) suggest to estimate the integral $\int f(x)\rho(x)\mathrm{d}x$ by $I_{\varpi,N}^{\mathrm{NEO}}(f) = N^{-1} \sum_{i=1}^N \sum_{k \in \mathbb{Z}} w_k(X^i)f(T^k(X^i))$ where $\{X^i\}_{i=1}^N$ are i.i.d. samples from the proposal $\rho$, which is denoted by $X^{1:N} \overset{\mathrm{iid}}{\sim} \rho$.

This estimator is obtained by a weighted combination of the elements of the independent forward and backward orbits $\{T^k(X^i)\}_{k \in \mathbb{Z}}$ with $X^{1:N} \overset{\mathrm{iid}}{\sim} \rho$. This estimator is referred to as NEO-IS. Choosing $f \equiv L$ provides the NEO-IS estimator of the normalizing constant of $\pi$:

---
**Algorithm 1** NEO-IS Sampler

1. Sample $X^{1:N} \overset{\mathrm{iid}}{\sim} \rho$ for $i \in [N]$.
2. For $i \in [N]$, compute the path $(T^j(X^i))_{j \in \mathbb{Z}}$ and weights $(w_j(X^i))_{j \in \mathbb{Z}}$.
3. $I_{\varpi,N}^{\mathrm{NEO}}(f) = N^{-1} \sum_{i=1}^N \sum_{k \in \mathbb{Z}} w_k(X^i)f(T^k(X^i))$.

---

$$\widehat{Z}_{X^i}^{\varpi} = \sum\nolimits_{k \in \mathbb{Z}} L(T^k(X^i))w_k(X^i) , \quad \widehat{Z}_{X^{1:N}}^{\varpi} = N^{-1} \sum\nolimits_{i=1}^N \widehat{Z}_{X^i}^{\varpi} . \tag{4}$$

We now study the performance of the NEO-IS estimator. The following two quantities play a fundamental role in the analysis:

$$E_T^{\varpi} = \mathbb{E}_{X \sim \rho}\big[\big(\sum\nolimits_{k \in \mathbb{Z}} w_k(X)L(T^k(X))/Z\big)^2\big], M_T^{\varpi} = \sup\nolimits_{x \in \mathbb{R}^d} \sum\nolimits_{k \in \mathbb{Z}} w_k(x)L(T^k(x))/Z . \tag{5}$$

**Theorem 1.** *$\widehat{Z}_{X^{1:N}}^{\varpi}$ is an unbiased estimator of Z. If $E_T^{\varpi} < \infty$, then, $\mathbb{E}[|\widehat{Z}_{X^{1:N}}^{\varpi}/Z - 1|^2] = N^{-1}(E_T^{\varpi} - 1)$. If $M_T^{\varpi} < \infty$, then, for any $\delta \in (0,1)$, with probability $1 - \delta$, $\sqrt{N}\,\big|\widehat{Z}_{X^{1:N}}^{\varpi}/Z - 1\big| \leq M_T^{\varpi}\sqrt{\log(2/\delta)/2}$.*

The (elementary) proof is postponed to Appendix A.3. $E_T^{\varpi}$ plays the role of the second-order moment of the importance weights $\mathbb{E}_{X \sim \rho}[L^2(X)]$ which is key to the performance of IS algorithms

[1, 2]. In addition, since the NEO-IS estimator $\widehat{Z}^{\varpi}_{X^{1:N}}$ is unbiased, the Cauchy–Schwarz inequality implies that $\mathbb{E}_{X\sim\rho}\big[\big(\sum_{k\in\mathbb{Z}} w_k(X)\mathrm{L}(\mathrm{T}^k(X)))^2\big] \geq \mathrm{Z}^2$ and hence that $E^{\varpi}_{\mathrm{T}} \geq 1$. Note that if $\|\mathrm{L}\|_\infty = \sup_{x\in\mathbb{R}^d} \mathrm{L}(x) < \infty$, then since the weights are uniformly bounded by $\Omega\varpi_0^{-1}$, we have $M^{\varpi}_{\mathrm{T}} \leq \|\mathrm{L}\|_\infty \Omega\varpi_0^{-1}/\mathrm{Z}$.

Using the NEO-IS estimate $\widehat{Z}^{\varpi}_{X^{1:N}}$ of the normalizing constant, we can construct a self-normalized IS estimate of $\int f(x)\pi(x)\mathrm{d}x$:

$$J^{\mathrm{NEO}}_{\varpi,N}(f) = N^{-1}\sum_{i=1}^{N} \frac{\widehat{Z}^{\varpi}_{X^i}}{\widehat{Z}^{\varpi}_{X^{1:N}}} \sum_{k\in\mathbb{Z}} \frac{\mathrm{L}(\mathrm{T}^k(X^i))w_k(X^i)}{\widehat{Z}^{\varpi}_{X^i}} f(\mathrm{T}^k(X^i)) , \qquad (6)$$

referred to as NEO-SNIS estimator. This expression may seem unnecessarily complicated but highlights the hierarchical structure of the estimator. We combine estimators $(\widehat{Z}^{\varpi}_{X^i})^{-1}\sum_{k\in\mathbb{Z}} \mathrm{L}(\mathrm{T}^k(X^i))w_k(X^i)f(\mathrm{T}^k(X^i))$ evaluated on the forward and backward orbits through the points $\{X^i\}_{i=1}^{N}$ using the normalized weights $\{\widehat{Z}^{\varpi}_{X^i}/\widehat{Z}^{\varpi}_{X^{1:N}}\}_{i=1}^{N}$. Although the NEO-IS estimator is unbiased, the NEO-SNIS is in general biased. However, for bounded functions, both the bias and the variance of the NEO-SNIS estimator are $O(N^{-1})$, with constants proportional to $E^{\varpi}_{\mathrm{T}}$. For $g$ a $\pi$-integrable function, we set $\pi(g) = \int g(x)\pi(x)\mathrm{d}x$.

**Theorem 2.** *Assume that $E^{\varpi}_{\mathrm{T}} < \infty$. Then, for any function $g$ satisfying $\sup_{x\in\mathbb{R}^d} |g(x)| \leq 1$ on $\mathbb{R}^d$, and $N \in \mathbb{N}$*

$$\mathbb{E}_{X^{1:N}\overset{\mathrm{iid}}{\sim}\rho}\big[|J^{\mathrm{NEO}}_{\varpi,N}(g) - \pi(g)|^2\big] \leq 4\cdot N^{-1}E^{\varpi}_{\mathrm{T}} , \qquad (7)$$

$$\Big|\mathbb{E}_{X^{1:N}\overset{\mathrm{iid}}{\sim}\rho}\big[J^{\mathrm{NEO}}_{\varpi,N}(g) - \pi(g)\big]\Big| \leq 2\cdot N^{-1}E^{\varpi}_{\mathrm{T}} . \qquad (8)$$

*If $M^{\varpi}_{\mathrm{T}} < \infty$, then for $\delta \in (0,1]$, with probability at least $1-\delta$,*

$$\sqrt{N}|J^{\mathrm{NEO}}_{\varpi,N}(g) - \pi(g)| \leq \|g\|_\infty M^{\varpi}_{\mathrm{T}}\sqrt{32\log(4/\delta)} . \qquad (9)$$

The proof is postponed to Appendix A.4. These results extend to NEO-SNIS estimators the results known for self-normalized IS estimators; see e.g., [1, 2] and the references therein. The upper bounds stated in this result suggest it is good practice to keep $E^{\varpi}_{\mathrm{T}}/N$ small in order to obtain sensible approximations. For two pdfs $p$ and $q$ on $\mathbb{R}^d$, denote by $\mathrm{D}_{\chi^2}(p,q) = \int\{p(x)/q(x) - 1\}^2 q(x)\mathrm{d}x$ the $\chi^2$-divergence between $p$ and $q$.

**Lemma 3.** *For any nonnegative sequence $(\varpi_k)_{k\in\mathbb{Z}}$, we have $E^{\varpi}_{\mathrm{T}} \leq \mathrm{D}_{\chi^2}(\pi\|\rho_{\mathrm{T}}) + 1$.*

The proof is postponed to Appendix A.5. Lemma 3 suggests that accurate sampling requires $N$ to scale linearly with the $\chi^2$-divergence between the target $\pi$ and the extended proposal $\rho_{\mathrm{T}}$.

**Remark 1.** We can extend NEO to non homogeneous flows, replacing the family $\{\mathrm{T}^k\colon k\in\mathbb{Z}\}$ with a collection of mappings $\{\mathrm{T}_k\colon k\in\mathbb{Z}\}$. This would allow us to consider further flexible classes of transformations such as normalizing flows; see e.g. [21]. The $\chi^2$-divergence $\mathrm{D}_{\chi^2}(\pi\|\rho_{\mathrm{T}})$ provides a natural criterion for learning the transformation. We leave this extension to future work.

**Conformal Hamiltonian transform** The efficiency of NEO relies heavily on the choice of T. Intuitively, a sensible choice of T requires that (i) $E^{\varpi}_{\mathrm{T}}$ is small, i.e. $\rho_{\mathrm{T}}$ should be close to $\pi$ by Lemma 3 (see (5)), (ii) the inverse $\mathrm{T}^{-1}$ and the Jacobian of T are easy to compute. Following [23], we use for T a discretization of a conformal Hamiltonian dynamics. Assume that $U(\cdot) = -\log\pi(\cdot)$ is continuously differentiable. We consider the augmented distribution $\tilde{\pi}(q,p) \propto \exp\{-U(q) - K(p)\}$ on $\mathbb{R}^{2d}$, where $q$ is the position, $p$ is the momentum, and $K(p) = p^T\mathrm{M}^{-1}p/2$ is the kinetic energy, with M a positive definite mass matrix. By construction, the marginal distribution of the momentum under $\tilde{\pi}$ is the target pdf $\pi(q) = \int \tilde{\pi}(q,p)\mathrm{d}p$. The conformal Hamiltonian ODE associated with $\tilde{\pi}$ is defined by

$$\mathrm{d}q_t/\mathrm{d}t = \nabla_p H(q_t,p_t) = \mathrm{M}^{-1}p_t , \qquad (10)$$
$$\mathrm{d}p_t/\mathrm{d}t = -\nabla_q H(q_t,p_t) - \gamma p_t = -\nabla U(q_t) - \gamma p_t ,$$

where $H(q,p) = U(q) + K(p)$, and $\gamma > 0$ is a damping constant. Any solution $(q_t,p_t)_{t\geq 0}$ of (10) satisfies setting $H_t = H(q_t,p_t)$, $\mathrm{d}H_t/\mathrm{d}t = -\gamma p_t^T\mathrm{M}^{-1}p_t \leq 0$. Hence, all orbits converge to fixed points that satisfy $\nabla U(q) = 0$ and $p = 0$; see e.g. [12, 17].

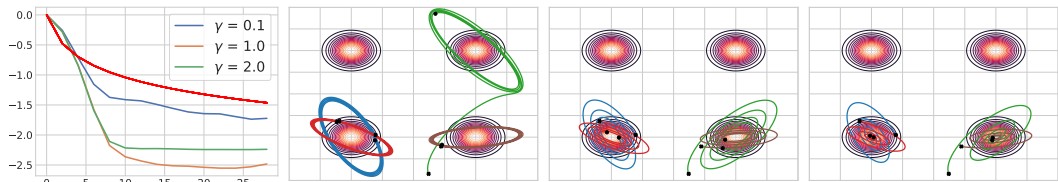

Figure 1: Left: $E_{\mathrm{T}_h}^{\mathbb{1}_{[K]}}(K) - 1$ vs $E^{\mathrm{IS}}(K) - 1$ (red) in $\log_{10}$-scale as a function of the length of trajectories $K$ (the lower the better). Second left to right: Four examples of orbits with the same random seed for different values of $\gamma$ (from left to right, $\gamma = 0.1, 1, 2$).

In the applications below, we consider the conformal version of the symplectic Euler (SE) method of (10), see [12]. This integrator can be constructed as a splitting of the two conformal and conservative parts of the system (10). When composing a dissipative with a symplectic operator, we set for all $(q, p) \in \mathbb{R}^{2d}$, $\mathrm{T}_h(q, p) = (q + h\mathrm{M}^{-1}\{\mathrm{e}^{-h\gamma}p - h\nabla U(q)\}, \mathrm{e}^{-h\gamma}p - h\nabla U(q))$, where $h > 0$ is a discretization stepsize. This transformation can be connected with classical momentum optimization schemes, see [12, Section 4]. By [12, Section 3], for any $h > 0$ $\mathrm{T}_h$ is a $\mathrm{C}^1$-diffeomorphism on $\mathbb{R}^{2d}$ with Jacobian given by $\mathbf{J}_{\mathrm{T}_h}(q, p) = \mathrm{e}^{-\gamma hd}$. In addition, its inverse is $\mathrm{T}_h^{-1}(q, p) = (q - h\mathrm{M}^{-1}p, \mathrm{e}^{\gamma h}\{p + h\nabla U(q - h\mathrm{M}^{-1}p)\})$. Therefore, the weight (3) of the NEO estimator is given by

$$w_k(q, p) = \frac{\varpi_k \tilde{\rho}(\mathrm{T}_h^k(q, p))\mathrm{e}^{-\gamma khd}}{\sum_{j \in \mathbb{Z}} \varpi_{k+j} \tilde{\rho}(\mathrm{T}_h^{-j}(q, p))\mathrm{e}^{\gamma jhd}} ,$$

where $\tilde{\rho}(q, p) \propto \rho(q)\mathrm{e}^{-K(p)}$. Figure 1 displays for different values of $\gamma$ on a log-scale the bound $E_{\mathrm{T}_h}^{\mathbb{1}_{[0:K]}} - 1$ appearing in Theorem 1 as a function of $K$, here we use the sequence of weights $(\varpi_k)_{k \in \mathbb{Z}} = (\mathbb{1}_{[0:K]}(k))_{k \in \mathbb{Z}}$ (i.e. only the $K + 1$ first elements of the forward orbits are used and are equally weighted). For comparison, we also present on the same plot the bounds achieved by averaging $K + 1$ independent IS estimates, $E^{\mathrm{IS}}(K) - 1 = (K + 1)^{-1}\mathbb{E}_{X \sim \rho}[\mathrm{L}(X)^2]$. Interestingly, Figure 1 shows that there is a trade-off in the choice of $\gamma$ which controls the exploration of the state space by the Hamiltonian dynamics since the higher $\gamma$, the faster the orbits converge towards the modes. This fast convergence prevents a "good" exploration of the space; e.g. $E_{\mathrm{T}_h}^{\mathbb{1}_{[0:K]}}$ is smaller for $\gamma = 1.0$ than for $\gamma = 2.0$ when $K > 7$.

## 3 NEO-MCMC algorithm

We now derive an MCMC method to sample from $\pi$ based on the NEO-IS estimator. A natural idea consists in adapting the Sampling Importance Resampling procedure (SIR) (see for example [24, 27]) to the NEO framework.

The SIR method to sample $J_{\varpi, N}^{\mathrm{NEO}}$ (see (6)) consists of 4 steps.

(SIR-1) Draw independently $X^{1:N} \overset{\mathrm{iid}}{\sim} \rho$ and compute the associated forward and backward orbits $\{\mathrm{T}^k(X^i)\}_{k \in \mathbb{Z}}$ of the point.

(SIR-2) Compute the normalizing constants associated with each orbit $\{\widehat{Z}_{X^i}^\varpi\}_{i=1}^N$.

(SIR-3) Sample an orbit index $I^N \in [N]$ with probability $\{\widehat{Z}_{X^i}^\varpi / \sum_{j=1}^N \widehat{Z}_{X^j}^\varpi\}_{i=1}^N$.

(SIR-4) Draw the iteration index $K^N$ on the $I^N$-th orbit with probability $\{\mathrm{L}(\mathrm{T}^k(X^{I^N}))w_k(X^{I^N})/\widehat{Z}_{X^{I^N}}^\varpi\}_{k \in \mathbb{Z}}$.

The resulting draw is denoted by $U^N = \mathrm{T}^{K^N}(X^{I^N})$. By construction, for any bounded function $f$, we get that $\mathbb{E}\left[f(U^N)|X^{1:N}, I^N\right] = \{\widehat{Z}_{X^{I^N}}^\varpi\}^{-1} \sum_{k \in \mathbb{Z}} w_k(X_{I^N})\mathrm{L}(\mathrm{T}^k(X_{I^N}))$ which implies $\mathbb{E}\left[f(U^N)|X^{1:N}\right] = J_{\varpi, N}^{\mathrm{NEO}}(f)$ (see (6)). Using Theorem 2, we therefore obtain $|\mathbb{E}[f(U^N)] - \int f(z)\pi(z)\mathrm{d}z| \leq 10^{1/2}\|f\|_\infty E_{\mathrm{T}}^\varpi N^{-1}$, showing that the law of the random variable $\mu_N = \mathrm{Law}(U^N)$ converges in total variation to $\pi$ as $N \to \infty$,

$$\|\mu_N - \pi\|_{\mathrm{TV}} = \sup_{\|f\|_\infty \leq 1} |\mu_N(f) - \pi(f)| \leq 10^{1/2} E_{\mathrm{T}}^\varpi N^{-1} . \tag{11}$$

Based on [3], we now derive the NEO-MCMC procedure, which in a nutshell consists in iterating the SIR procedure while keeping a conditioning point (or equivalently, orbit); see Appendix C. The convergence of NEO-MCMC does not rely on letting $N \to \infty$: the NEO-MCMC works as soon as $N \geq 2$, although as we will see below the mixing time decreases as $N$ increases.

This procedure is summarized in Algorithm 2. The NEO-MCMC procedure is an iterated algorithm which produces a sequence $\{(Y_n, U_n)\}_{n \in \mathbb{N}}$ of points in $\mathbb{R}^d$. The $n$-th iteration of the NEO-MCMC algorithm consists in two main steps: 1) updating the conditioning point $Y_{n-1} \to Y_n$

---

**Algorithm 2** NEO-MCMC Sampler

At step $n \in \mathbb{N}^*$, given the conditioning orbit point $Y_{n-1}$.

**Step 1: Update the conditioning point**

1. Set $X_n^1 = Y_{n-1}$ and for any $i \in \{2, \ldots, N\}$, sample $X_n^i \overset{\text{iid}}{\sim} \rho$.
2. Sample the orbit index $I_n$ with probability proportional to $(\widehat{Z}_{X_n^i}^{\varpi})_{i \in [N]}$, (4).
3. Set $Y_n = X_n^{I_n}$.

**Step 2: Output a sample**

4. Sample index $K_n$ with probability proportional to $\{w_k(Y_n) \mathrm{L}(\mathrm{T}^k(Y_n))/\widehat{Z}_{Y_n}^{\varpi}\}_{k \in \mathbb{Z}}$
5. Output $U_n = \mathrm{T}^{K_n}(Y_n)$.

---

2) sampling $U_n$ by selecting a point in the orbit $\{\mathrm{T}^k(Y_n)\}_{k \in \mathbb{Z}}$ of the conditioning point. Compared to SIR, only the generation of the points (step (SIR-1)) is modified: we set $X_n^1 = Y_{n-1}$ (the **conditioning point**), and then draw $X_n^{2:N} \overset{\text{iid}}{\sim} \rho$.

The sequence $\{Y_n\}_{n \in \mathbb{N}}$ defined by Algorithm 2 is a Markov chain: $\mathbb{P}(Y_n \in \mathsf{A} \mid Y_{0:n-1}) = \mathbb{P}(Y_n \in \mathsf{A} \mid Y_{n-1}) = P(Y_n, \mathsf{A})$ where

$$P(y, \mathsf{A}) = \int \delta_y(\mathrm{d}x^1) \prod_{j=2}^N \rho(x^j)\mathrm{d}x^j \sum_{i=1}^N \frac{\widehat{Z}_{x^i}^{\varpi}}{\sum_{j=1}^N \widehat{Z}_{x^j}^{\varpi}} \mathbb{1}_{\mathsf{A}}(x^i) , \quad y \in \mathbb{R}^d , \mathsf{A} \in \mathcal{B}(\mathbb{R}^d) . \quad (12)$$

Note that this Markov kernel describes the way, at stage $n+1$, the conditioning point $Y_{n+1}$ is selected given $Y_n$, which **depends only on** the estimator of the normalizing constants associated with each orbit, **but not** on the sample $U_n$ selected on the conditioning orbit. In addition, given the conditioning point $Y_n$ at the $n$-th iteration, the conditional distribution of the output sample $U_n$ is $\mathbb{P}(U_n \in \mathsf{B} \mid I_n, X_n^{1:N}) = \mathbb{P}(U_n \in \mathsf{B} \mid Y_n) = Q(Y_n, \mathsf{B})$ where

$$Q(y, \mathsf{B}) = \sum_{k \in \mathbb{Z}} \frac{w_k(y)\mathrm{L}(\mathrm{T}^k(y))}{\widehat{Z}_y^{\varpi}} \mathbb{1}_{\mathsf{B}}(\mathrm{T}^k(y)) , \quad y \in \mathbb{R}^d , \mathsf{B} \in \mathcal{B}(\mathbb{R}^d) . \quad (13)$$

With these notations, if the Markov chain is started at $Y_0 = y$, then for any $n \in \mathbb{N}$, the law of the $n$-th conditioning point is $\mathbb{P}(Y_n \in \mathsf{A} \mid Y_0 = y) = P^n(y, \mathsf{A})$ and the law of the $n$-th sample is $\mathbb{P}(U_n \in \mathsf{B} \mid Y_0) = P^n Q(y, \mathsf{B})$. Define $\tilde{\pi}$ the pdf given for $y \in \mathbb{R}^d$ by

$$\tilde{\pi}(y) = \frac{\rho(y)}{\mathrm{Z}} \sum_{k \in \mathbb{Z}} w_k(y)\mathrm{L}(\mathrm{T}^k(y)) = \frac{\rho(y)\widehat{Z}_y^{\varpi}}{\mathrm{Z}} . \quad (14)$$

The following theorem shows that, for any initial condition $y \in \mathbb{R}^d$, the distribution of the variable $Y_n$ converges in total variation to $\tilde{\pi}$ and that the distribution of $U_n$ converges to $\pi$.

**Theorem 4.** *The Markov kernel $P$ is reversible with respect to the distribution $\tilde{\pi}$, ergodic and Harris positive, i.e., for all $y \in \mathbb{R}^d$, $\lim_{n \to \infty} \|P^n(y, \cdot) - \tilde{\pi}\|_{\mathrm{TV}} = 0$. In addition, $\pi = \tilde{\pi}Q$ and $\lim_{n \to \infty} \|P^n Q(y, \cdot) - \pi\|_{\mathrm{TV}} = 0$. Moreover, for any bounded function $g$ and any $y \in \mathbb{R}^d$, $\lim_{n \to \infty} n^{-1} \sum_{i=0}^{n-1} g(U_i) = \pi(g)$, $\mathbb{P}$-almost surely, where $\{U_i\}_{i \in \mathbb{N}}$ is defined in Algorithm 2 with $Y_0 = y$.*

The proof is postponed to Appendix A.6.

**Remark 2.** We may provide another sampling procedure of $\{Y_n\}_{n \in \mathbb{N}}$. Define the pdf on the extended space $[N] \times \mathbb{R}^{dN}$ by $\check{\pi}(i, x^{1:N}) = N^{-1}\tilde{\pi}(x^i) \prod_{j=1, j \neq i}^N \rho(x^j)$. Consider a Gibbs sampler targeting $\check{\pi}$ consisting in (a) sampling $X_n^{1:N \setminus \{I_{n-1}\}} \mid (I_{n-1}, X_{n-1}) \sim \prod_{j \neq I_{n-1}} \rho(x^j)$, (b) sampling $I_n \mid X_n^{1:N} \sim \mathrm{Cat}(\{\widehat{Z}_{X_n^i}^{\varpi} / \sum_{j=1}^N \widehat{Z}_{X_n^j}^{\varpi}\}_{i=1}^N)$ and (c) set $Y_n = X_n^{I_n}$. This algorithm is a Gibbs sampler on $\check{\pi}$ and we easily verify that the distribution of $\{Y_n\}_{n \in \mathbb{N}}$ is the same as Algorithm 2.

The next theorem provides non asymptotic quantitative bounds on the convergence in total variation. The main interest of NEO-MCMC algorithm is motivated empirically from observed behaviour: the mixing time of the corresponding Markov chain improves as $N$ increases. This behaviour is quantified theoretically in the next theorem. Moreover, this improvement is obtained with little extra computational overhead, since sampling $N$ points from the proposal distribution $\rho$, computing the forward and backward orbits of the points and evaluating the normalizing constants $\{\widehat{Z}_{X_n^i}^{\varpi}\}_{i=1}^N$ can be performed in parallel.

**Theorem 5.** *Assume that $M_{\mathrm{T}}^{\varpi} < \infty$, see (5). Set $\epsilon_N = (N-1)/(2M_{\mathrm{T}}^{\varpi} + N - 2)$ and $\kappa_N = 1 - \epsilon_N$. Then, for any $y \in \mathbb{R}^d$ and $k \in \mathbb{N}$, $\|P^k(y, \cdot) - \tilde{\pi}\|_{\mathrm{TV}} \le \kappa_N^k$ and $\|P^k Q(y, \cdot) - \pi\|_{\mathrm{TV}} \le \kappa_N^k$.*

Instead of sampling the new points $X_n^{2:N}$ independently from $\rho$ (Step 1 in Algorithm 2), it is possible to draw the proposals $X_n^{1:N}$ conditional to the current point $Y_{n-1}$; see [29, 8, 26, 25] for related works. Following [25], we use a reversible Markov kernel with respect to the proposal $\rho$, i.e., such that $\rho(x)m(x, x') = \rho(x')m(x', x)$, assuming for simplicity that this kernel has density $m(x, x')$. If $\rho = \mathrm{N}(0, \sigma^2 \mathrm{Id}_d)$, an appropriate choice is an autoregressive kernel $m(x, x') = \mathrm{N}(x'; \alpha x, \sigma^2(1 - \alpha^2) \mathrm{Id}_d)$. More generally, we can use a Metropolis–Hastings kernel with invariant distribution $\rho$. In this case, $r_1(x^1, x^{1:N \setminus \{1\}}) = \prod_{j=2}^N m(x^{j-1}, x^j)$ and for each $i \in [2 : N]$,

$$r_i(x^i, x^{1:N \setminus \{i\}}) = \prod_{j=1}^{i-1} m(x^{j+1}, x^j) \prod_{j=i+1}^N m(x^{j-1}, x^j) . \tag{15}$$

Since $m$ is reversible with respect to $\rho$, for all $i, j \in [N]$, $\rho(x^i)r_i(x^i, x^{1:N \setminus \{i\}}) = \rho(x^j)r_j(x^j, x^{1:N \setminus \{j\}})$ where $r_i(x^i; x^{1:N \setminus \{i\}})$ defines the the conditional distribution of $X^{1:N \setminus \{i\}}$ given $X^i = x^i$. The only modification in Algorithm 2 is Step 1, which is replaced by: *Draw $U_n \in [N]$ uniformly, set $X_n^{U_n} = Y_{n-1}$ and sample $X_n^{1:N \setminus \{U_n\}} \sim r_{U_n}(X_n^{U_n}, \cdot)$.* The validity of this procedure is established in Appendix A.6.

## 4   Continuous-time version of NEO and NEIS

The NEO framework can be thought of as an extension of NEIS introduced in [23]. NEIS focuses on normalizing constant estimation and should be therefore compared with NEO-IS. In [23], the authors do not consider possible extensions of these ideas to sampling problems. We consider here how NEO could be adapted to continuous-time dynamical system. Proofs of the statements and detailed technical conditions are postponed to Appendix B.

Consider the Ordinary Differential Equation (ODE) $\dot{x}_t = b(x_t)$ , where $b \colon \mathbb{R}^d \to \mathbb{R}^d$ is a smooth vector field. Denote by $(\phi_t)_{t \in \mathbb{R}}$ the flow of this ODE (assumed to be well-behaved). Under appropriate regularity condition $\mathbf{J}_{\phi_t}(x) = \exp(\int_0^t \nabla \cdot b(\phi_s(x))\mathrm{d}s)$; see Lemma S5. Let $\varpi : \mathbb{R} \to \mathbb{R}_+$ be a nonnegative smooth function with finite support, with $\Omega^c = \int_{-\infty}^{\infty} \varpi(t)\mathrm{d}t$. The continuous-time counterpart of the proposal distribution (1) is $\rho_{\mathrm{T}}^c(x) = (\Omega^c)^{-1} \int_{-\infty}^{\infty} \varpi(t)\rho(\phi_{-t}(x))\mathbf{J}_{\phi_{-t}}(x)\mathrm{d}t$, which is a continuous mixture of the pushforward of the proposal $\rho$ by the flow of $(\phi_s)_{s \in \mathbb{R}}$. Assuming for simplicity that $\rho(x) > 0$ for all $x \in \mathbb{R}^d$, then $\rho_{\mathrm{T}}^c(x) > 0$ for all $x \in \mathbb{R}^d$, and using again the IS formula, for any nonnegative function $f$,

$$\int f(x)\rho(x)\mathrm{d}x = \int f(x)\frac{\rho(x)}{\rho_{\mathrm{T}}^c(x)}\rho_{\mathrm{T}}^c(x)\mathrm{d}x = \int \left[\int_{-\infty}^{\infty} w_t^c(x)f(\phi_t(x))\mathrm{d}t\right]\rho(x)\mathrm{d}x, \tag{16}$$

$$w_t^c(x) = \varpi(t)\rho(\phi_t(x))\mathbf{J}_{\phi_t}(x) \Big/ \int_{-\infty}^{\infty} \varpi(s+t)\rho(\phi_s(x))\mathbf{J}_{\phi_s}(x)\mathrm{d}s . \tag{17}$$

These relations are the continuous-time counterparts of (2). Eqs. (16)-(17) define a version of NEIS [23], with a finite support weight function $\varpi$; see Appendices B.2 and B.3 for weight functions with infinite support. This identity is of theoretical interest but must be discretized to obtain a computationally tractable estimator. For $h > 0$, denote by $\mathrm{T}_h$ an integrator with stepsize $h > 0$ of the ODE $\dot{x} = b(x)$. We may construct NEO-IS and NEO-SNIS estimators based on the transform $\mathrm{T} \leftarrow \mathrm{T}_h$ and weights $\varpi_k \leftarrow \varpi(kh)$. We might show that for any bounded function $f$ and for any $x \in \mathbb{R}^d$, $\lim_{h \downarrow 0} \sum_{k \in \mathbb{Z}} w_k(x)f(\mathrm{T}_h^k(x)) = \int_{-\infty}^{\infty} w_t^c(x)f(\phi_t(x))\mathrm{d}t$, where we omitted here the

dependency in $h$ of $w_k$. Therefore, taking $h \downarrow 0^+$, the NEO-IS converges to the continuous-version (16)-(17). There is however an important difference between NEO and the NEIS method in [23] which stems from the way (16)-(17) are discretized. Compared to NEIS, NEO-IS using $T \leftarrow T_h$ and weights $\varpi_k \leftarrow \varpi(kh)$ is unbiased for any stepsize $h > 0$. NEIS uses an approach inspired by the nested-sampling approach, which amounts to discretizing the integral in (16) also in the state-variable $x$; see [28, 7]. This discretization is biased which prevents the use of this approach to develop MCMC sampling algorithm; see Appendix B.

## 5   Experiments and Applications

**Normalizing constant estimation**   The performance of NEO-IS is assessed on different normalizing constant estimation benchmarks; see [13]. We focus on two challenging examples. Additional experiments and discussion on hyperparameter choice are given in the supplementary material, see Appendix D.1.

**(1) Mixture of Gaussian** (MG25): $\pi(x) = P^{-1} \sum_{i=1}^{P} \mathrm{N}(x; \mu_{i,j}, D_d)$, where $d \in \{10, 20, 45\}$, $D_d = \mathrm{diag}(0.01, 0.01, 0.1, \ldots, 0.1)$ and $\mu_{i,j} = [i, j, 0, \ldots, 0]^T$ with $i, j \in \{-2, \ldots, 2\}$.

**(2) Funnel distribution** (Fun) $\pi(x) = \mathrm{N}(x_1; 0, a^2) \prod_{i=1}^{d} \mathrm{N}(x_i; 0, \mathrm{e}^{2bx_1})$ with $d \in \{10, 20, 45\}$, $a = 1$, and $b = 0.5$. In both case, the proposal is $\rho = \mathrm{N}(0, \sigma_\rho^2 \, \mathrm{Id}_d)$ with $\sigma_\rho^2 = 5$.

The NEO-IS estimator is compared with (i) the IS estimator using the proposal $\rho$, (ii) the Adaptive Importance Sampling (AIS) estimator of [30], (iii) Stochastic Normalizing Flows (SNF)[1] and (iv) the Neural Importance Sampling (NIS)[2]. For NEO-IS, we use $\varpi_k = \mathbb{1}_{[K]}(k)$ with $K = 10$ (ten steps on the forward orbit), and conformal Hamiltonian dynamics $\gamma = 1$, $M = 5 \cdot \mathrm{Id}_d$ for dimensions $d = \{10, 20\}$, and $\gamma = 2.5$ for $d = 45$ (where $\gamma$ is the damping factor, $M$ the mass matrix, $h$ is the stepsize of the integrator). The parameters of AIS are set to obtain a complexity comparable to NEO-IS; see Appendix D.1. For NIS, we use the default parameters and for SNF we used the same architectures as in [33]. In Fun, we set $\gamma = 0.2$, $K = 10$, $M = 5 \cdot \mathrm{Id}_d$, and $h = 0.3$. The IS estimator was based on $5 \cdot 10^5$ samples, and NIS, NEO-IS and AIS were computed with $5 \cdot 10^4$ samples. Figure 2 shows that NEO-IS consistently outperforms the competing methods.

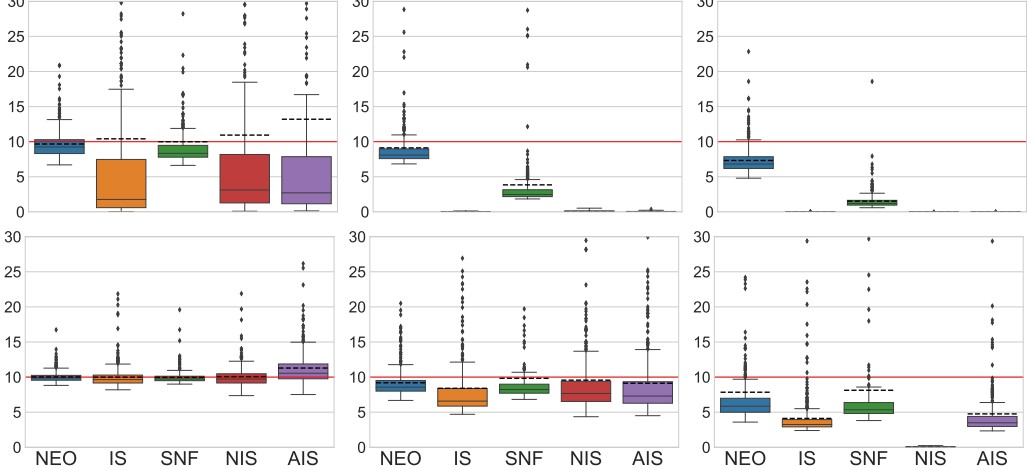

Figure 2:   Boxplots of 500 independent estimations of the normalizing constant in dimension $d = \{10, 20, 45\}$ (from left to right) for MG25 (top) and Fun (bottom). The true value is given by the red line. The figure displays the median (solid lines), the interquartile range, and the mean (dashed lines) over the 500 runs.

---

[1]Implementation available at https://github.com/noegroup/stochastic_normalizing_flows.
[2]Implementation available at https://github.com/ndeutschmann/zunis.

**Sampling** NEO-MCMC is assessed for the distributions (`MG25`) ($d = 40$) and `Fun` ($d = 20$). NEO-MCMC sampler is compared with (i) the No-U-Turn Sampler - Pyro library [4] - and (ii) i-SIR algorithm [25]. The proposal distribution is $\rho = \mathrm{N}(0, \sigma_\rho^2 \mathrm{Id}_d)$ with $\sigma_\rho^2 = 5$. Dependent proposals are used (see (15)) with $m(x, x') = \mathrm{N}(x'; \alpha x, \sigma_\rho^2 (1 - \alpha^2) \mathrm{Id}_d)$ with $\alpha = 0.99$. For NUTS, the default parameters are used. For i-SIR, we use the same number of proposals $N = 10$, proposal distribution and dependent proposal as for NEO-MCMC. To perform a fair comparison, we use the same clock time for all three algorithms. The number of iterations for correlated i-SIR, NEO-MCMC, and NUTS are $n = 4 \cdot 10^6$, $n = 4 \cdot 10^5$, and $n = 5 \cdot 10^5$, respectively. Figure 3 displays the empirical

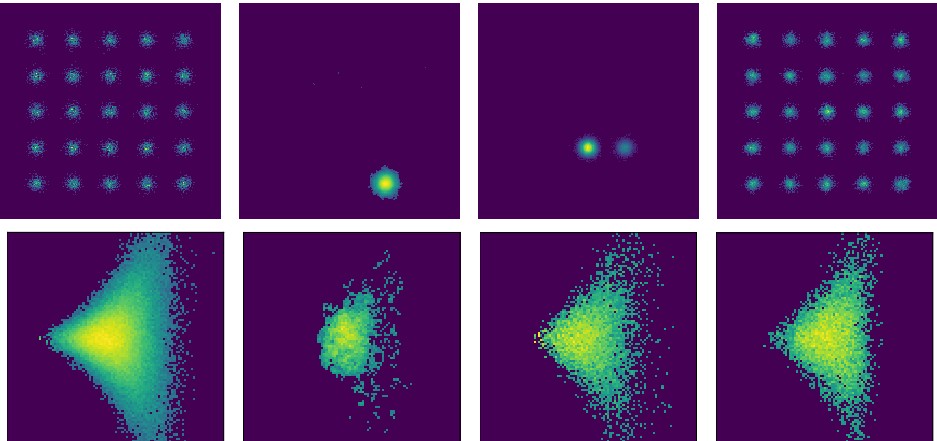

Figure 3: Empirical 2-D histogram of the samples of different algorithms targeting `MG25` (top) and `Fun` (bottom). Left to right: samples from the target distribution, correlated i-SIR, NUTS, NEO-MCMC.

two-dimensional histograms of the two first coordinates of samples from the ground truth, i-SIR, NUTS and NEO-MCMC sampler. It is worthwhile to note that NEO-MCMC algorithm performs much better for `MG25` which is a very challenging distribution, even for SOTA algorithm such as NUTS, which struggles to cross energy barriers between modes. For `Fun`, NEO-MCMC performs favourably with respect to NUTS, which is well adapted for this type of distributions.

**Block Gibbs Inpainting with Deep Generative models and** NEO-**MCMC** We apply NEO-MCMC to the task of sampling the posterior of a deep latent variable model. To be consistent with the rest of the paper, we use non-standard notation here with $x$ being the latent variable and $z$ the observation. More precisely, we assume that $x \sim \mathrm{N}(0, \mathrm{Id}_d)$ and a conditional distribution $p(z \mid x)$ which generates an image $z = (z^1, \dots, z^D) \in \mathbb{R}^D$. Given a family of parametric *decoders* $\{x \mapsto p_\theta(z \mid x), \theta \in \Theta\}$, and a training set $\mathcal{D} = \{z_i\}_{i=1}^M$, training involves finding the MLE $\theta^* = \arg\max_{\theta \in \Theta} p_\theta(\mathcal{D})$. As $p_\theta(z) = \int p_\theta(z \mid x) p(x) \mathrm{d}x$, the likelihood is intractable and to alleviate this problem, [14] proposed to train jointly an approximate posterior $q_\phi(x|z)$ that maximizes a tractable lower-bound on the log-likelihood: $\mathrm{ELBO}(z, \theta, \phi) = \mathbb{E}_{X \sim q_\phi(\cdot|z)}[\log p_\theta(z, X)/q_\phi(X|z)] \leq p_\theta(z)$, where $q_\phi(x \mid z)$ is a tractable conditional distribution with parameters $\phi \in \Phi$. It is assumed in the sequel that conditional to the latent variable $x$, the coordinates are independent, *i.e.* $p_\theta(z \mid x) = \prod_{i=1}^D p_\theta(z^i|x)$.

Note that it is possible to train VAE with the NEO algorithm, using the unbiased estimate of the normalizing constant to construct an ELBO. This approach is described in the supplement Appendix E. We do not focus on this approach here and assume that the VAE has been trained and we are only interested in the sampling problem. In our experiment, we use a VAE trained on CelebA dataset [3] [16]. We consider the Block Gibbs inpainting task introduced in [15, Section 5.2.2]. Given an image $z$, denote by $[z^t, z^b]$ the top and the bottom half pixels. Assume only $z_\star^t$ is observed, then we are interested in in-painting the bottom of an image by the posterior distribution of $z^b$ given $z_\star^t$. This is achieved using Block Gibbs sampling. A two-stage Gibbs sampler amounts to (a) sampling $p_{\theta*}(x|z^t, z^b)$ and (b) sampling $p_{\theta*}(z^b|x, z^t) = p_{\theta*}(z^b|x)$ (since $z^b$ and $z^t$ are independent conditional on $x$). Given $z_k = (z_\star^t, z_k^b)$, we sample at each step $x_k \sim p_{\theta*}(x \mid z_k)$ and then

---

[3]Publicly available online, see https://github.com/YannDubs/disentangling-vae/tree/master/results/betaH_celeba

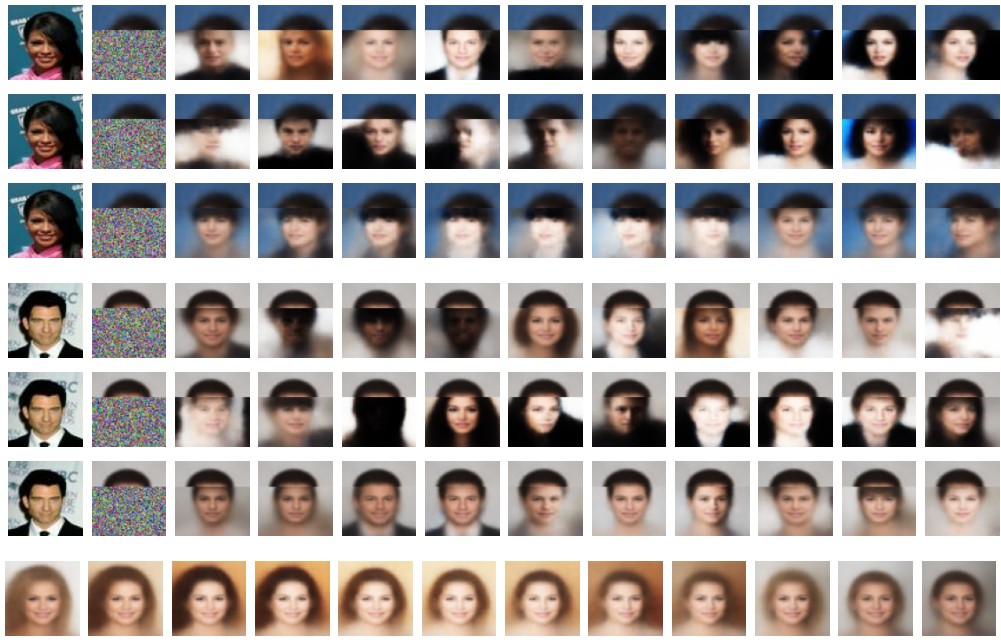

Figure 4: Two examples for the Gibbs inpainting task for CelebA dataset. From top to bottom (twice) : i-SIR, HMC and NEO-MCMC: From left to right, original image, blurred image to reconstruct, and output every 5 iterations of the Markov chain. Last line: a forward orbit used in NEO-MCMC for the second example.

$z_{k+1}^b \sim p_{\theta*}(z^b \mid x_k)$. We then set $z_{k+1} = (z_*^t, z_{k+1}^b)$. Stage (b) is elementary but stage (a) is challenging. We use an MCMC-within-Gibbs scheme using different samplers. We use the following decomposition of $p_{\theta*}(x \mid z) \propto \rho(x)\mathrm{L}(x)$ for $\rho(x) \propto q_{\phi*}^\beta(x \mid z)$ and $\mathrm{L}(x) = p_{\theta*}(x, z)/q_{\phi*}^\beta(x \mid z)$ with $\beta \in (0, 1)$. It is possible to sample from $\rho(x)$ as $q_{\phi*}(x \mid z)$ is Gaussian. In our experiments with CelebA and the chosen trained VAE, we have $x \in \mathbb{R}^{10}$ (recall that $x$ is our latent variable here), $z \in \mathbb{R}^{12288}$, and use $\beta = 0.1$. We then compare i-SIR, HMC and NEO-MCMC sampler in stage (a), with the same computational complexity ($N = 10$, $K = 12$, $\gamma = 0.2$ for NEO-MCMC, $N = 120$ for i-SIR, and HMC is run with $K = 20$ leap-frog steps). Again, NEO-MCMC and i-SIR use dependent proposals, with $m$ a Random Walk Metropolis kernel with stepsize 0.1. For each algorithm, 10 steps are performed. Figure 4 displays the evolution of the resulting Markov chains. The samples clearly illustrate that NEO-MCMC mixes better than i-SIR and HMC. More details and examples are presented in the supplementary.

## 6   Conclusion

In this paper, we have proposed a new family of algorithms, NEO, for computing normalizing constants and sampling from complex distributions. This methodology comes with asymptotic and non-asymptotic convergence guarantees. For normalizing constant estimation, NEO-IS compares favorably to state-of-the-art algorithms on difficult benchmarks. NEO-MCMC is also able to sample some complex distributions: it is particularly well-adapted to sampling multimodal distributions, thanks to its proposal mechanism which avoids being trapped in local modes. There are numerous potential extensions to this work. For example, it would be interesting to consider deterministic transformations other than conformal Hamiltonian dynamics integrators. These transformations could be trained, as for Neural IS, using a variation lower bound. It would also be interesting to further investigate the influence of the mixture weights $\{\varpi_k\}_{k \in \mathbb{Z}}$ on the efficiency of NEO.

**Broader impact:** Sampling from complex target distributions and computing their normalizing constants has numerous applications. Our work proposes novel methods to address such problems and has thus potential applications in many areas. This work does not present any foreseeable societal consequence.

## Acknowledgments and Disclosure of Funding

Arnaud Doucet is partly supported by the EPSRC grant EP/R034710/1. He also acknowledges support of the UK Defence Science and Technology Laboratory (DSTL) and EPSRC under grant EP/R013616/1. This is part of the collaboration between US DOD, UK MOD and UK EPSRC under the Multidisciplinary University Research Initiative. Alain Durmus and Eric Moulines acknowledge support of the Lagrange Mathematical and Computing Research Center.

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
