# A  Proofs

## A.1  Additional notation

By abuse of notation, we denote by $\rho$ and $\tilde{\pi}$ the probability measures with density with respect to the Lebesgue measure $\rho$ and $\tilde{\pi}$ respectively.

## A.2  Proof of (3)

The second expression of $w_k$ follows from $\mathbf{J}_{\mathrm{T}^{-j}}(\mathrm{T}^k(x)) = \mathbf{J}_{\mathrm{T}^{k-j}}(x)/\mathbf{J}_{\mathrm{T}^k}(x)$ which implies

$$w_k(x) = \varpi_k \rho(\mathrm{T}^k(x))\Big/\sum\nolimits_{j\in\mathbb{Z}} \varpi_j \rho(\mathrm{T}^{k-j}(x))\mathbf{J}_{\mathrm{T}^{-j}}(\mathrm{T}^k(x)) \ ,$$

$$= \varpi_k \rho(\mathrm{T}^k(x))\mathbf{J}_{\mathrm{T}^k}(x)\Big/\sum\nolimits_{j\in\mathbb{Z}} \varpi_j \rho(\mathrm{T}^{k-j}(x))\mathbf{J}_{\mathrm{T}^{k-j}}(x) = \varpi_k \rho_{-k}(x)\Big/\sum\nolimits_{i\in\mathbb{Z}} \varpi_{k+i}\rho_i(x) \ \ .$$

## A.3  Proof of Theorem 1

The unbiasedness of $\widehat{\mathrm{Z}}^{\varpi}_{X^{1:N}}$ follows directly from (2). Moreover, as $\widehat{\mathrm{Z}}^{\varpi}_{X^{1:N}}$ is unbiased and $E^{\varpi}_{\mathrm{T}} < \infty$, we can write

$$\mathrm{Var}_\rho[\widehat{\mathrm{Z}}^{\varpi}_X/\,\mathrm{Z}] = \mathbb{E}_\rho[(\widehat{\mathrm{Z}}^{\varpi}_X/\,\mathrm{Z})^2] - 1 = E^{\varpi}_{\mathrm{T}} - 1 \ . \tag{S1}$$

As $X^{1:N} \overset{\mathrm{iid}}{\sim} \rho$, $\mathrm{Var}_\rho[\widehat{\mathrm{Z}}^{\varpi}_{X^{1:N}}/\,\mathrm{Z}] = N^{-1}\,\mathrm{Var}_\rho[\widehat{\mathrm{Z}}^{\varpi}_X/\,\mathrm{Z}]$. Finally, if $M^{\varpi}_{\mathrm{T}} < \infty$, then Hoeffding's inequality applies and we can write for any $\epsilon > 0$,

$$\mathbb{P}(|\widehat{\mathrm{Z}}^{\varpi}_{X^{1:N}}/\,\mathrm{Z} - 1| > \epsilon) \le 2\exp(-2N\epsilon^2/(M^{\varpi}_{\mathrm{T}})^2) \ . \tag{S2}$$

Writing $\delta = 2\exp(-2N\epsilon^2/(M^{\varpi}_{\mathrm{T}})^2)$, we identify $\log(2/\delta) = 2N\epsilon^2/(M^{\varpi}_{\mathrm{T}})^2$ and $\epsilon = M^{\varpi}_{\mathrm{T}}\sqrt{\log(2/\delta)/(2N)}$. Plugging this expression of $\epsilon$ in (S2) concludes the proof.

## A.4  Proof of Theorem 2

We first present two auxiliary lemmas necessary to establish Theorem 2.

**Lemma S1.** *Let $A, B$ be two integrable random variables satisfying $|A/B| \le M$ almost surely and denote $a = \mathbb{E}[A]$, $b = \mathbb{E}[B]$. Then,*

$$|\mathbb{E}[A/B] - a/b| \le \frac{\sqrt{\mathrm{Var}(A/B)\,\mathrm{Var}(B)}}{b} \ , \tag{S3}$$

$$\mathrm{Var}(A/B) \le \mathbb{E}\left[|A/B - a/b|^2\right] \le \frac{2}{B^2}\left(\mathbb{E}\left[|A_N - A|^2\right] + M^2\mathbb{E}\left[|B_N - B|^2\right]\right) \ . \tag{S4}$$

*Proof.* Write first, using the Cauchy-Schwarz inequality,

$$\left|\mathbb{E}\left[\frac{A}{B}\right] - \frac{a}{b}\right| = \left|\mathbb{E}\left[\frac{A}{B}\right] - \frac{\mathbb{E}[A]}{b}\right| = \left|\mathbb{E}\left[A\left(\frac{1}{B} - \frac{1}{b}\right)\right]\right| \ ,$$

$$= \left|\mathbb{E}\left[\frac{A}{B}\left(\frac{b-B}{b}\right)\right]\right| = \left|\mathbb{E}\left[\left(\frac{A}{B} - \mathbb{E}\left[\frac{A}{B}\right]\right)\left(\frac{B-b}{b}\right)\right]\right| \ ,$$

$$\le \frac{\sqrt{\mathrm{Var}(A/B)}\sqrt{\mathrm{Var}(B)}}{b} \ .$$

Moreover, using $|A/B| \le M$ yields

$$\left|\frac{A}{B} - \frac{a}{b}\right| = \left|\frac{1}{b}(A - a) + A\left(\frac{1}{B} - \frac{1}{b}\right)\right| \le \frac{1}{b}|A - a| + \frac{|A|}{Bb}|B - b| \ ,$$

$$\le \frac{1}{b}|A - a| + \frac{M}{b}|B - b| \ .$$

Therefore,

$$|A/B - a/b|^2 \le \frac{2}{b^2}\left(|A - a|^2 + M^2|B - b|^2\right) \ ,$$

Using that $\mathbb{E}\left[|A/B - a/b|^2\right] = \mathrm{Var}(A/B) + |\mathbb{E}[A/B] - a/b|^2$ concludes the proof. $\qquad\square$

21  We get the following lemma from [7, Lemma 4].

22  **Lemma S2.** *Assume that $A$ and $B$ are random variables and that there exist positive constants*
23  *$b, M, C, K$ such that*

24      *(i) $|A/B| \leq M$, $\mathbb{P}$-a.s.,*

25      *(ii) for all $\epsilon > 0$ and all $N \geq 1$, $\mathbb{P}\left(|B - b| > \epsilon\right) \leq K \exp(-R\epsilon^2)$,*

26      *(iii) for all $\epsilon > 0$ and all $N \geq 1$, $\mathbb{P}\left(|A| > \epsilon\right) \leq K \exp\left(-R\epsilon^2/M^2\right)$,*

*then,*
$$\mathbb{P}(|A/B| \geq \epsilon) \leq 2K \exp(-Rb^2\epsilon^2/4M^2) \ .$$

27  *Proof.* By the triangle inequality,
$$|A/B| = \left| \frac{A}{B}(b - B)b^{-1} + b^{-1}A \right| \ ,$$
$$\leq b^{-1}|A/B||b - B| + b^{-1}|A| \leq Mb^{-1}|b - B| + b^{-1}|A| \ .$$

28  Therefore,
$$\{|A/B| \geq \epsilon\} \subseteq \left\{ |B - b| \geq \frac{\epsilon b}{2M} \right\} \cup \left\{ |A| \geq \frac{\epsilon b}{2} \right\} \ .$$

29  Then, conditions (ii) and (iii) imply that
$$\mathbb{P}\left(|A/B| \geq \epsilon\right) \leq \mathbb{P}\left(|B - b| \geq \frac{\epsilon b}{2M}\right) + \mathbb{P}\left(|A| \geq \frac{\epsilon b}{2}\right) \ ,$$
$$\leq 2K \exp(-Rb^2\epsilon^2/(4M^2)) \ .$$

30  $\square$

31  *Proof of Theorem 2.* Let $g : \mathbb{R}^d \to \mathbb{R}$ such that $\sup_{x \in \mathbb{R}^d} |g|(x) \leq 1$ and denote $\pi(g) = \int g \mathrm{d}\pi$. We
32  use Lemma S1 with $A = A_N$ and $B = \widehat{Z}^{\varpi}_{X^{1:N}}$ where
$$A_N = \frac{1}{N} \sum_{i=1}^N \sum_{k \in \mathbb{Z}} w_k(X^i) \mathrm{L}(\mathrm{T}^k(X^i))g(\mathrm{T}^k(X^i)) \ , \quad \widehat{Z}^{\varpi}_{X^{1:N}} = \frac{1}{N} \sum_{i=1}^N \sum_{k \in \mathbb{Z}} w_k(X^i) \mathrm{L}(\mathrm{T}^k(X^i)) \ .$$
(S5)
33  By construction, since $\sup_{x \in \mathbb{R}^d} |g|(x) \leq 1$, almost surely $A_N/\widehat{Z}^{\varpi}_{X^{1:N}} \leq 1$ and $\mathrm{Var}(\widehat{Z}^{\varpi}_{X^{1:N}}) =$
34  $N^{-1}\mathrm{Var}(\widehat{Z}^{\varpi}_{X^1})$. Then, using (2) with $a = \mathbb{E}[A_N] = Z\,\pi(g)$ and $b = \mathbb{E}[\widehat{Z}^{\varpi}_{X^{1:N}}] = Z$, Lemma S1
35  implies
$$\left| J^{\mathrm{NEO}}_{\varpi,N}(g) - \pi(g) \right| = \left| \mathbb{E}[A_N/\widehat{Z}^{\varpi}_{X^{1:N}}] - a/b \right| \leq N^{-1/2} \sqrt{\mathrm{Var}(A_N/\widehat{Z}^{\varpi}_{X^{1:N}})\mathrm{Var}(\widehat{Z}^{\varpi}_{X^1})} \ . \quad (S6)$$

36  On the other hand,
$$\mathbb{E}\left[|A_N - a|^2\right] = N^{-1}\mathbb{E}_{X \sim \rho}\left[\left\{ \sum_{k \in \mathbb{Z}} w_k(X)\mathrm{L}(\mathrm{T}^k(X))g(\mathrm{T}^k(X)) - Z\,\pi(g) \right\}^2\right] \leq N^{-1} Z^2 E^{\varpi}_{\mathrm{T}} \ .$$

37  These inequalities yield using $\mathrm{Var}(\widehat{Z}^{\varpi}_{X^1}) \leq E^{\varpi}_{\mathrm{T}}$ and Lemma S1 again:
$$\mathbb{E}\left[|J^{\mathrm{NEO}}_{\varpi,N}(g) - \pi(g)|^2\right] \leq \frac{2}{N}(E^{\varpi}_{\mathrm{T}} + \mathrm{Var}(\widehat{Z}^{\varpi}_{X^1})) \leq \frac{4}{N}E^{\varpi}_{\mathrm{T}} \ ,$$
$$|\mathbb{E}\left[J^{\mathrm{NEO}}_{\varpi,N}(g) - \pi(g)\right]| \leq \frac{\sqrt{2(E^{\varpi}_{\mathrm{T}} + \mathrm{Var}(\widehat{Z}^{\varpi}_{X^1}))\mathrm{Var}(\widehat{Z}^{\varpi}_{X^1})}}{N} \leq \frac{2E^{\varpi}_{\mathrm{T}}}{N} \ ,$$

38  which concludes the proof.

39  Define
$$\tilde{A}_N = N^{-1} \sum_{i=1}^N \sum_{k \in \mathbb{Z}} w_k(X^i)\mathrm{L}(\mathrm{T}^k(X^i))\left( g(\mathrm{T}^k(X^i)) - \pi(g) \right) \ .$$

With this notation, the proof of (9) relies on the application of Lemma S2 to $A = \tilde{A}_N$ and $B = \widehat{Z}^{\varpi}_{X^{1:N}}$, since

$$J^{\mathrm{NEO}}_{\varpi,N}(g) - \pi(g) = A_N/\widehat{Z}^{\varpi}_{X^{1:N}} \ .$$

As $\sup_{x \in \mathbb{R}^d} |g|(x) \leq 1$, we get that $\tilde{A}_N/\widehat{Z}^{\varpi}_{X^{1:N}} \leq 2$. By (2), $\mathbb{E}[\widehat{Z}^{\varpi}_{X^{1:N}}] = \mathrm{Z}$ and $\widehat{Z}^{\varpi}_{X^{1:N}} = N^{-1} \sum_{i=1}^{N} W_i$ with $W_i = \sum_{k \in \mathbb{Z}} w_k(X^i) \mathrm{L}(\mathrm{T}^k(X^i)) \leq M^{\varpi}_{\mathrm{T}}$. Then, by Hoeffding's inequality, for all $\varepsilon > 0$,

$$\mathbb{P}(|B_N - \mathrm{Z}| > \varepsilon) \leq 2\exp(-2N(\varepsilon/M^{\varpi}_{\mathrm{T}})^2) \ .$$

Similarly, $A_N$ is centered and $A_N = N^{-1} \sum_{i=1}^{N} U_i$ with

$$U_i = \sum_{k \in \mathbb{Z}} w_k(X^i) \mathrm{L}(\mathrm{T}^k(X^i))\{g(\mathrm{T}^k(X^i)) - \pi(g)\}$$

and $|U_i| \leq 2M^{\varpi}_{\mathrm{T}}$ almost surely. By Hoeffding's inequality, for all $\varepsilon > 0$,

$$\mathbb{P}(|A_N| > \epsilon) \leq 2\exp(-N\varepsilon^2/(8(M^{\varpi}_{\mathrm{T}})^2)) \ .$$

The assumptions of Lemma S2 are met so that

$$\mathbb{P}(|J^{\mathrm{NEO}}_{\varpi,N}(g) - \pi(g)| > \varepsilon) \leq 4\exp(-\varepsilon^2 N \, \mathrm{Z}^2/[32(M^{\varpi}_{\mathrm{T}})^2]) \ ,$$

which concludes the proof. $\qquad\square$

## A.5  Proof of Lemma 3

As $w_k(x) = \varpi_k \rho(\mathrm{T}^k(x))/\{\Omega \rho_{\mathrm{T}}(\mathrm{T}^k(x))\}$, by Jensen's inequality,

$$E^{\varpi}_{\mathrm{T}} = \int \left(\sum_{k \in \mathbb{Z}} w_k(x) \mathrm{L}(\mathrm{T}^k(x))/\mathrm{Z}\right)^2 \rho(x)\mathrm{d}x = \int \left(\sum_{k \in \mathbb{Z}} \frac{\varpi_k}{\Omega} \frac{\pi(\mathrm{T}^k(x))}{\rho_{\mathrm{T}}(\mathrm{T}^k(x))}\right)^2 \rho(x)\mathrm{d}x \ ,$$

$$\leq \int \sum_{k \in \mathbb{Z}} \frac{\varpi_k}{\Omega} \left(\frac{\pi(\mathrm{T}^k(x))}{\rho_{\mathrm{T}}(\mathrm{T}^k(x))}\right)^2 \rho(x)\mathrm{d}x \ ,$$

$$\leq \Omega^{-1} \sum_{k \in \mathbb{Z}} \varpi_k \int \left(\frac{\pi(\mathrm{T}^k(x))}{\rho_{\mathrm{T}}(\mathrm{T}^k(x))}\right)^2 \rho(x)\mathrm{d}x \ .$$

Using the change of variables $y = \mathrm{T}^k(x)$ yields, by (1),

$$E^{\varpi}_{\mathrm{T}} \leq \Omega^{-1} \sum_{k \in \mathbb{Z}} \varpi_k \int \left(\frac{\pi(y)}{\rho_{\mathrm{T}}(y)}\right)^2 \rho(\mathrm{T}^{-k}(y)) \mathbf{J}_{\mathrm{T}^{-k}}(y)\mathrm{d}y \leq \int \left(\frac{\pi(y)}{\rho_{\mathrm{T}}(y)}\right)^2 \rho_{\mathrm{T}}(y)\mathrm{d}y \ .$$

## A.6  Proofs of NEO MCMC sampler

*Proof of Theorem 4.* Note first that by symmetry, we have

$$P(y, \mathsf{A}) = N^{-1} \int \sum_{i=1}^{N} \delta_y(\mathrm{d}x^i) \prod_{j=1, j \neq i}^{N} \rho(x^j)\mathrm{d}x^j \sum_{k=1}^{N} \frac{\widehat{Z}^{\varpi}_{x^k}}{\sum_{j=1}^{N} \widehat{Z}^{\varpi}_{x^j}} \mathbb{1}_{\mathsf{A}}(x^k) \ . \qquad (\mathrm{S7})$$

We begin with the proof of reversibility of $P$ with respect to $\tilde{\pi}$. Let $f, g$ be nonnegative measurable functions. By definition of $P$,

$$\int \tilde{\pi}(\mathrm{d}y) P(y, \mathrm{d}y') f(y) g(y') = \frac{1}{N\,\mathrm{Z}} \int \sum_{i=1}^{N} \rho(\mathrm{d}y) \widehat{Z}^{\varpi}_y f(y) \delta_y(\mathrm{d}x^i) \prod_{l=1, l \neq i}^{N} \rho(\mathrm{d}x^l) \sum_{k=1}^{N} \frac{\widehat{Z}^{\varpi}_{x^k}}{\sum_{j=1}^{N} \widehat{Z}^{\varpi}_{x^j}} g(x^k) \ ,$$

$$= \frac{1}{N\,\mathrm{Z}} \int \sum_{i=1}^{N} \widehat{Z}^{\varpi}_{x^i} f(x^i) \prod_{l=1}^{N} \rho(\mathrm{d}x^l) \sum_{k=1}^{N} \frac{\widehat{Z}^{\varpi}_{x^k}}{\sum_{j=1}^{N} \widehat{Z}^{\varpi}_{x^j}} g(x^k) \ ,$$

$$= \frac{1}{N\,\mathrm{Z}} \int \prod_{l=1}^{N} \rho(\mathrm{d}x^l) \frac{\sum_{i=1}^{N} \widehat{Z}^{\varpi}_{x^i} f(x^i) \sum_{k=1}^{N} \widehat{Z}^{\varpi}_{x^k} g(x^k)}{\sum_{j=1}^{N} \widehat{Z}^{\varpi}_{x^j}} \ ,$$

$$= \int \tilde{\pi}(\mathrm{d}y) P(y, \mathrm{d}y') f(y') g(y) \ ,$$

47 which shows that $P$ is $\tilde{\pi}$-reversible. We now establish that $P$ is $\tilde{\pi}$-irreducible. We have for $y \in \mathbb{R}^d$,
48 $\mathsf{A} \in \mathcal{B}(\mathbb{R}^d)$,

$$
\begin{aligned}
P(y, \mathsf{A}) &= \int \delta_y(\mathrm{d}x^1) \sum_{i=1}^N \frac{\widehat{Z}_{x^i}^{\varpi}}{N\widehat{Z}_{x^{1:N}}^{\varpi}} \mathbb{1}_\mathsf{A}(x^i) \prod_{j=2}^N \rho(\mathrm{d}x^j) \\
&= \int \frac{\widehat{Z}_y^{\varpi}}{\widehat{Z}_y^{\varpi} + \sum_{j=2}^N \widehat{Z}_{x^j}^{\varpi}} \mathbb{1}_\mathsf{A}(x) \prod_{j=2}^N \rho(\mathrm{d}x^j) + \int \sum_{i=2}^N \frac{\widehat{Z}_{x^i}^{\varpi}}{\widehat{Z}_y^{\varpi} + \sum_{j=2}^N \widehat{Z}_{x^j}^{\varpi}} \mathbb{1}_\mathsf{A}(x^i) \prod_{j=2}^N \rho(\mathrm{d}x^j) \\
&\geq \sum_{i=2}^N \int \frac{\widehat{Z}_{x^i}^{\varpi}}{\widehat{Z}_y^{\varpi} + \widehat{Z}_{x^i}^{\varpi} + \sum_{j=2,j\neq i}^N \widehat{Z}_{x^j}^{\varpi}} \mathbb{1}_\mathsf{A}(x^i) \prod_{j=2}^N \rho(\mathrm{d}x^j) \\
&\geq \sum_{i=2}^N \int \tilde{\pi}(\mathrm{d}x^i) \mathbb{1}_\mathsf{A}(x^i) \int \frac{\mathsf{Z}}{\widehat{Z}_y^{\varpi} + \widehat{Z}_{x^i}^{\varpi} + \sum_{j=2,j\neq i}^N \widehat{Z}_{x^j}^{\varpi}} \prod_{j=2,j\neq i}^N \rho(\mathrm{d}x^j) .
\end{aligned}
$$

49 Since the function $f \colon z \mapsto (z+a)^{-1}$ is convex on $\mathbb{R}_+$ for $a > 0$, we get for $i \in \{2, \ldots, N\}$,

$$
\begin{aligned}
\int \frac{\mathsf{Z}}{\widehat{Z}_y^{\varpi} + \widehat{Z}_{x^i}^{\varpi} + \sum_{j=2,j\neq i}^N \widehat{Z}_{x^j}^{\varpi}} \prod_{j=2,j\neq i}^N \rho(\mathrm{d}x^j) &\geq \frac{\mathsf{Z}}{\widehat{Z}_y^{\varpi} + \widehat{Z}_{x^i}^{\varpi} + \int \sum_{j=2,j\neq i}^N \widehat{Z}_{x^j}^{\varpi} \prod_{j=2,j\neq i}^N \rho(\mathrm{d}x^j)} \\
&\geq \frac{\mathsf{Z}}{\widehat{Z}_y^{\varpi} + \widehat{Z}_{x^i}^{\varpi} + \mathsf{Z}(N-2)} . \quad \text{(S8)}
\end{aligned}
$$

50 Therefore, for $\mathsf{A} \in \mathcal{B}(\mathbb{R}^d)$ satisfying $\tilde{\pi}(\mathsf{A}) > 0$, we get $P(y, \mathsf{A}) > 0$ for any $y \in \mathbb{R}^d$ since $\widehat{Z}_x^{\varpi} < \infty$
51 for any $x \in \mathbb{R}^d$. By definition, $P$ is $\tilde{\pi}$-irreducible.

52 We show that $P$ is Harris recurrent using [19, Corollary 2]. To this end, since $P$ is $\tilde{\pi}$-
53 irreducible, it is sufficient to show that $P$ is a Metropolis type kernel. Define $\alpha(x^1, x^2) = (N -$
54 $1) \int \prod_{j=3}^N \rho(\mathrm{d}x^j) \widehat{Z}_{x^2}^{\varpi} / \sum_{j=1}^N \widehat{Z}_{x^j}^{\varpi}$ for $x^1, x^2 \in \mathbb{R}^d$ and $\rho_{2:N}(\mathrm{d}x^{2:N}) = \{\prod_{j=2}^N \rho_{2:N}(x^j)\}\mathrm{d}x^{2:N}$.
55 Then, by (12), we get with this notation, for $y \in \mathbb{R}^d$, $\mathsf{A} \in \mathcal{B}(\mathbb{R}^d)$,

$$
\begin{aligned}
&P(y, \mathsf{A}) \\
&= \int \delta_y(\mathrm{d}x^1) \rho_{2:N}(\mathrm{d}x^{2:N}) \sum_{i=2}^N \frac{\widehat{Z}_{x^i}^{\varpi}}{N\widehat{Z}_{x^{1:N}}^{\varpi}} \mathbb{1}_\mathsf{A}(x^i) + \int \delta_y(\mathrm{d}x^1) \rho_{2:N}(\mathrm{d}x^{2:N}) \frac{\widehat{Z}_{x^1}^{\varpi}}{N\widehat{Z}_{x^{1:N}}^{\varpi}} \mathbb{1}_\mathsf{A}(x^1) \\
&= \sum_{i=2}^N \int \delta_y(\mathrm{d}x^1) \rho_{2:N}(\mathrm{d}x^{2:N}) \frac{\widehat{Z}_{x^i}^{\varpi}}{N\widehat{Z}_{x^{1:N}}^{\varpi}} \mathbb{1}_\mathsf{A}(x^i) + \int \delta_y(\mathrm{d}x^1) \rho_{2:N}(\mathrm{d}x^{2:N}) \frac{\widehat{Z}_{x^1}^{\varpi}}{N\widehat{Z}_{x^{1:N}}^{\varpi}} \mathbb{1}_\mathsf{A}(x^1) \\
&= \sum_{i=2}^N \int \delta_y(\mathrm{d}x^1) \rho(\mathrm{d}x^i) \int \prod_{j=2,j\neq i}^N \rho(x^j) \mathrm{d}x^j \frac{\widehat{Z}_{x^i}^{\varpi} \mathbb{1}_\mathsf{A}(x^i)}{N\widehat{Z}_{x^{1:N}}^{\varpi}} + \int \delta_y(\mathrm{d}x^1) \rho_{2:N}(\mathrm{d}x^{2:N}) \frac{\widehat{Z}_{x^1}^{\varpi} \mathbb{1}_\mathsf{A}(x^1)}{N\widehat{Z}_{x^{1:N}}^{\varpi}} \\
&= \sum_{i=2}^N \int \frac{\alpha(y, x^i)}{(N-1)} \mathbb{1}_\mathsf{A}(x^i) \rho(\mathrm{d}x^i) + \int \delta_y(\mathrm{d}x^1) \rho_{2:N}(\mathrm{d}x^{2:N}) \left\{ 1 - \sum_{i=2}^N \frac{\widehat{Z}_{x^i}^{\varpi}}{N\widehat{Z}_{x^{1:N}}^{\varpi}} \right\} \mathbb{1}_\mathsf{A}(x^1) \\
&= \int_\mathsf{A} \alpha(y, y') \rho(y') \mathrm{d}y' + \left( 1 - \int \alpha(y, y') \rho(y') \mathrm{d}y' \right) \delta_y(\mathsf{A}) . \quad \text{(S9)}
\end{aligned}
$$

56 With the terminology of [19, Corollary 2], $P$ is Metropolis type kernel and therefore is Harris
57 recurrent.

58 Note that Algorithm 2 defines a Markov chain $\{Y_i, U_i\}_{i \in \mathbb{N}}$ taking for $U_0$ an arbitrary initial point
59 with Markov kernel denoted by $\tilde{P}$. By abuse of notation, we denote by $\{Y_i, U_i\}_{i \in \mathbb{N}}$ the canonical
60 process on the canonical space $(\mathbb{R}^d \times \mathbb{R}^d)^{\mathbb{N}}$ endowed with the corresponding $\sigma$-field and denote
61 by $\mathbb{P}_{y,u}$ the distribution associated with the Markov chain with kernel $\tilde{P}$ and initial distribution
62 $\delta_y \otimes \delta_u$. Denote for any $y \in \mathbb{R}^d$ by $\mathbb{P}_y$ the marginal distribution of $\mathbb{P}_{y,u}$ with respect to $\{Y_i\}_{i \in \mathbb{N}}$,
63 *i.e.* $\mathbb{P}_y(\mathsf{A}) = \mathbb{P}_{(y,u)}(\{Y_i\}_{i \in \mathbb{N}} \in \mathsf{A})$ for $u \in \mathbb{R}^d$, noting that by definition, $\mathbb{P}_{(y,u)}(\mathsf{A} \times (\mathbb{R}^d)^{\mathbb{N}})$ does not

64 depend on $u$. In addition, under $\mathbb{P}_y$, $\{Y_i\}_{i\in\mathbb{N}}$ is a Markov chain associated with $P$. Therefore, since
65 $P$ is $\tilde{\pi}$-irreducible and Harris recurrent, we get by [8, Theorem 11.3.1] and [19, Theorem 2, 3] for
66 any $y \in \mathbb{R}^d$, $\lim_{k\to\infty} \|\delta_y P^k - \tilde{\pi}\|_{\mathrm{TV}} = 0$ and for any bounded and measurable function $g$,

$$n^{-1}\sum_{k=1}^{n} g(Y_k) = \tilde{\pi}(g)\,, \qquad \mathbb{P}_y\text{-almost surely}\,. \tag{S10}$$

67 We now turn to proving the properties regarding $Q$. For any $\mathsf{B} \in \mathcal{B}(\mathbb{R}^d)$, using (2), we obtain

$$\int \tilde{\pi}(y)Q(y,\mathsf{B})\mathrm{d}y = \mathrm{Z}^{-1}\int \rho(y)\sum_{k\in\mathbb{Z}} w_k(y)\mathrm{L}(\mathrm{T}^k(y))\mathbb{1}_{\mathsf{B}}(\mathrm{T}^k(y))\mathrm{d}y = \pi(\mathsf{B})\,.$$

68 Using for all $y \in \mathbb{R}^d$, $\lim_{n\to\infty}\|P^n(y,\cdot) - \tilde{\pi}\|_{\mathrm{TV}} = 0$, we get $\lim_{n\to\infty}\|P^nQ(y,\cdot) - \pi\|_{\mathrm{TV}} = 0$. It
69 remains to show the stated Law of Large Numbers. Let $y, u \in \mathbb{R}^d$ and $g$ be a bounded measurable
70 function. Define for any $i \in \mathbb{N}^*$, $\tilde{U}_i = g(U_i) - Qg(Y_i)$. By definition, for any $i \in \mathbb{N}^*$, $\left|\tilde{U}_i\right| \leq$
71 $2\sup_{x\in\mathbb{R}^d}|g(x)|$ and $\mathbb{E}_{(y,u)}[\tilde{U}_i|\mathcal{F}_{i-1}] = 0$, where $\{\mathcal{F}_k\}_{k\in\mathbb{N}}$ is the canonical filtration. Therefore,
72 $\{\tilde{U}_i\}_{i\in\mathbb{N}^*}$ are $\{\mathcal{F}_k\}_{k\in\mathbb{N}}$-martingale increments and $\{S_k = \sum_{i=1}^{k}\tilde{U}_i\}_{k\in\mathbb{N}}$ is a $\{\mathcal{F}_k\}_{k\in\mathbb{N}}$-martingale.
73 Using [10, Theorem 2.18], we get

$$\lim_{n\to\infty}\{S_n/n\} = 0\,, \quad \mathbb{P}_{(y,u)}\text{-almost surely}\,. \tag{S11}$$

74 The proof is completed using that $\lim_{n\to\infty}\{n^{-1}\sum_{i=1}^{n} Qg(Y_i)\} = \tilde{\pi}(Qg) = \pi(g)$, $\mathbb{P}_y$-almost surely
75 by (S10) and therefore by definition, $\mathbb{P}_{(y,u)}$-almost surely. $\qquad\square$

76 *Proof of Theorem 5.* We have for $(x,\mathsf{A}) \in \mathbb{R}^d \times \mathcal{B}(\mathbb{R}^d)$,

$$P(y,\mathsf{A}) \geq \sum_{i=2}^{N}\int \tilde{\pi}(\mathrm{d}x^i)\mathbb{1}_{\mathsf{A}}(x^i)\int \frac{\mathrm{Z}}{\widehat{\mathrm{Z}}_y^{\varpi} + \widehat{\mathrm{Z}}_{x^i}^{\varpi} + \sum_{j=2,j\neq i}^{N}\widehat{\mathrm{Z}}_{x^j}^{\varpi}}\prod_{j=2,j\neq i}^{N}\rho(\mathrm{d}x^j)\,.$$

77 Moreover, as for any $x \in \mathbb{R}^d$, $\widehat{\mathrm{Z}}_x^{\varpi}/\mathrm{Z} \leq M_{\mathrm{T}}^{\varpi}$,

$$\int \frac{\mathrm{Z}}{\widehat{\mathrm{Z}}_y^{\varpi} + \widehat{\mathrm{Z}}_{x^i}^{\varpi} + \sum_{j=2,j\neq i}^{N}\widehat{\mathrm{Z}}_{x^j}^{\varpi}}\prod_{j=2,j\neq i}^{N}\rho(\mathrm{d}x^j) \geq \frac{\mathrm{Z}}{\widehat{\mathrm{Z}}_y^{\varpi} + \widehat{\mathrm{Z}}_{x^i}^{\varpi} + \mathrm{Z}(N-2)} \geq \frac{1}{2M_{\mathrm{T}}^{\varpi} + N - 2}\,.$$

78 We finally obtain the inequality

$$P(x,\mathsf{A}) \geq \tilde{\pi}(\mathsf{A}) \times \frac{N-1}{2M_{\mathrm{T}}^{\varpi} + N - 2} = \epsilon_N\tilde{\pi}(\mathsf{A})\,. \tag{S12}$$

79 The proof for $P$ is concluded from [8, Theorem 18.2.4].

80 As $\|P^k(y,\cdot) - \tilde{\pi}\|_{\mathrm{TV}} \leq \kappa_N^k$, for any bounded function $f$, $\|f\|_\infty \leq 1$, we have $|P^kf(y) - \tilde{\pi}(f)| \leq \kappa_N^k$,
81 by definition of the Total Variation Distance. Then, writing $f = Qg$ for any bounded function $g$,
82 $\|g\|_\infty \leq 1$, we have $\|f\|_\infty \leq 1$ and

$$|P^kf(y) - \tilde{\pi}(f)| = |P^kQg(y) - \tilde{\pi}Q(g)| = |P^kQg(y) - \pi(g)| \leq \kappa_N^k\,. \tag{S13}$$

83 $\qquad\qquad\qquad\qquad\qquad\qquad\qquad\qquad\qquad\qquad\qquad\qquad\qquad\qquad\qquad\qquad\qquad\square$

84 Write now $P$ the Markov kernel extending to correlated proposals: for $y \in \mathbb{R}^d$ and $\mathsf{A} \in \mathcal{B}(\mathbb{R}^d)$,

$$P(y,\mathsf{A}) = N^{-1}\int \sum_{i=1}^{N}\delta_y(\mathrm{d}x^i)r_i(x^i,\mathrm{d}x^{1:n\setminus\{i\}})\sum_{k=1}^{N}\frac{\widehat{\mathrm{Z}}_{x^k}^{\varpi}}{N\widehat{\mathrm{Z}}_{x^{1:N}}^{\varpi}}\mathbb{1}_{\mathsf{A}}(x^k)\,, \tag{S14}$$

85 where the Markov kernels $R_i$ are defined by $R_i(x^i,\mathrm{d}x^{1:N\setminus\{i\}}) = r_i(x^i,x^{1:N\setminus\{i\}})\mathrm{d}x^{1:N\setminus\{i\}}$ and $r_i$
86 by (15).

87 **Theorem S3.** *$P$ is $\tilde{\pi}$-invariant.*

*Proof.* Define the $Nd$-dimensional probability measure $\bar{\rho}_N(\mathrm{d}x^{1:N}) = \rho(\mathrm{d}x^1)R_1(x^1, \mathrm{d}x^{2:n})$. Let $\mathsf{A} \in \mathcal{B}(\mathbb{R}^d)$. Then, we have

$$
\begin{aligned}
\tilde{\pi}P(\mathsf{A}) &= N^{-1} \int \tilde{\pi}(\mathrm{d}y) \int \sum_{i=1}^{N} \delta_y(\mathrm{d}x^i) R_i(x^i, \mathrm{d}x^{1:n\backslash\{i\}}) \sum_{k=1}^{N} \frac{\widehat{Z}_{x^k}^{\varpi}}{N\widehat{Z}_{x^{1:N}}^{\varpi}} \mathbb{1}_{\mathsf{A}}(x^k) \\
&= (N\,\mathrm{Z})^{-1} \int \sum_{i=1}^{N} \rho(\mathrm{d}x^i) \widehat{Z}_{x^i}^{\varpi} R_i(x^i, \mathrm{d}x^{1:n\backslash\{i\}}) \sum_{k=1}^{N} \frac{\widehat{Z}_{x^k}^{\varpi}}{N\widehat{Z}_{x^{1:N}}^{\varpi}} \mathbb{1}_{\mathsf{A}}(x^k) \\
&= (N\,\mathrm{Z})^{-1} \int \bar{\rho}_N(\mathrm{d}x^{1:N}) \sum_{i=1}^{N} \widehat{Z}_{x^i}^{\varpi} \sum_{k=1}^{N} \frac{\widehat{Z}_{x^k}^{\varpi}}{N\widehat{Z}_{x^{1:N}}^{\varpi}} \mathbb{1}_{\mathsf{A}}(x^k) \\
&= (N\,\mathrm{Z})^{-1} \int \sum_{k=1}^{N} \widehat{Z}_{x^k}^{\varpi} \bar{\rho}_N(\mathrm{d}x^{1:N}) \mathbb{1}_{\mathsf{A}}(x^k) \\
&= (N\,\mathrm{Z})^{-1} \int \sum_{k=1}^{N} \widehat{Z}_{x^k}^{\varpi} \rho(\mathrm{d}x^k) \mathbb{1}_{\mathsf{A}}(x^k) = \tilde{\pi}(\mathsf{A}) \ .
\end{aligned}
$$

$\square$

# B   Continuous-time limit of NEO and NEIS

## B.1   Proof for the continuous-time limit

Consider $\bar{h} > 0$ and a family $\{\mathrm{T}_h \ : \ h \in \left(0, \bar{h}\right]\}$ of $\mathrm{C}^1$-diffeomorphisms. For $N \in \mathbb{N}^*$ and a bounded and continuous $f : \mathbb{R}^d \to \mathbb{R}$, write

$$
I_{\varpi,N,h}^{\mathrm{NEO}}(f) = N^{-1} \sum_{i=1}^{N} \sum_{k\in\mathbb{Z}} w_{k,h}(X^i) f(\mathrm{T}_h^k(X^i)) \ , \tag{S15}
$$

where $\{X_i\}_{i=1}^{N} \overset{\mathrm{iid}}{\sim} \rho$ and for some weight function $\varpi^{\mathrm{c}} : \mathbb{R} \to \mathbb{R}_+$ with bounded support (see **H**3), $k \in \mathbb{Z}$ and $h > 0$, setting $\varpi_{k,h} = \varpi^{\mathrm{c}}(kh)$,

$$
w_{k,h}(x) = \varpi_{k,h}\rho_{-k}(x) \Big/ \sum_{i\in\mathbb{Z}} \varpi_{k+i,h}\rho_i(x) \ . \tag{S16}
$$

We show in this section the convergence of the sequence of NEO-IS estimators $\{I_{\varpi,N,h}^{\mathrm{NEO}}(f) \ : \ h \in \left(0, \bar{h}\right]\}$ as $h \downarrow 0$ to its continuous counterpart, the version (16) of NEIS [16], with weight function $\varpi$, in the case where for any $h \in \left(0, \bar{h}\right]$, $\mathrm{T}_h$ corresponds to one step of a discretization scheme with stepsize $h$ of the ODE

$$
\dot{x}_t = b(x_t) \ , \tag{S17}
$$

where $b : \mathbb{R}^d \to \mathbb{R}^d$ is a drift function. We are particularly interested in the case where (S17) corresponds to the conformal Hamilonian dynamics (10) and $\{\mathrm{T}_h \ : \ h \in \left(0, \bar{h}\right]\}$ to its conformal symplectic Euler discretization: for all $(q, p) \in \mathbb{R}^{2d}$,

$$
\mathrm{T}_h(q, p) = (q + h\mathrm{M}^{-1}\{\mathrm{e}^{-h\gamma}p - h\nabla U(q)\}, \mathrm{e}^{-h\gamma}p - h\nabla U(q)) \ . \tag{S18}
$$

We make the following conditions on $b$, $\rho$, $\varpi^{\mathrm{c}}$ and $\{\mathrm{T}_h \ : \ h \in \left(0, \bar{h}\right]\}$.

**H1.** *The function $b$ is continuously differentiable and $L_b$-Lipschitz.*

Under **H**1, consider $(\phi_t)_{t\geq 0}$ the differential flow associated with (S17), *i.e.* $\phi_t(x) = x_t$ where $(x_t)_{t\in\mathbb{R}}$ is the solution of (S17) starting from $x$. Note that **H**1 implies that $(t, x) \mapsto \phi_t(x)$ is continuously differentiable on $\mathbb{R} \times \mathbb{R}^d$, see [11, Theorem 4.1 Chapter V].

**H**1 is satisfied in the case of the conformal Hamiltonian dynamics if the potential $U$ is continuously differentiable and with Lipschitz gradient, that is there exists $L_U \in \mathbb{R}_+^*$ such that for any $x_1, x_2 \in \mathbb{R}^d$, $\|\nabla U(x_1) - \nabla U(x_2)\| \leq L_U\|x_1 - x_2\|$.

112   **H2.** *For any $h \in \left(0, \bar{h}\right]$, $\mathrm{T}_h : \mathbb{R}^d \to \mathbb{R}^d$ is a $\mathrm{C}^1$-diffeomorphism. In addition, it holds:*

*(i) there exist $C \geq 0$ and $\delta \in (0, 1]$ such that for any $x \in \mathbb{R}^d$,*

$$\| \mathrm{T}_h(x) - (x + hb(x))\| \leq Ch^{1+\delta}(1 + \|x\|) \; ;$$

*(ii) for any $x \in \mathbb{R}^d$ and $T \in \mathbb{R}_+^*$,*

$$\lim_{h \downarrow 0} \max_{k \in [-\lfloor T/h \rfloor : \lfloor T/h \rfloor]} \|\mathbf{J}_{\phi_{kh}}(x) - \mathbf{J}_{\mathrm{T}_h^k}(x)\| = 0 \; .$$

113    Note that **H2** is automatically satisfied for the conformal symplectic Euler discretization (S18)
114    of the conformal Hamiltonian dynamics. Indeed, in that case $\operatorname{div} b(\phi_t(x)) = \gamma d$, and therefore
115    $\mathbf{J}_{\phi_t}(x) = e^{\gamma dt}$ for $t \in \mathbb{R}$, and for any $h > 0, k \in \mathbb{Z}$, $\mathbf{J}_{\mathrm{T}_h^k}(x) = e^{\gamma dhk}$; see [9].

116    Define

$$\operatorname{support}(\varpi^{\mathrm{c}}) = \{t \in \mathbb{R} \; : \; \varpi^{\mathrm{c}}(t) \neq 0\} \; . \tag{S19}$$

117   **H3.** *(i) $\rho$ is continuous and positive on $\mathbb{R}^d$*

*(ii) $\varpi^{\mathrm{c}}$ is piecewise continuous on $\mathbb{R}$, its support $\operatorname{support}(\varpi^{\mathrm{c}})$ is bounded and $\sup_{(s,t) \in \mathsf{A}_\varpi} \varpi^{\mathrm{c}}(t)/\varpi^{\mathrm{c}}(t + s) = m < \infty$ where*

$$\mathsf{A}_\varpi = \{(s, t) \in \mathbb{R}^2; \; t \in \operatorname{support}(\varpi^{\mathrm{c}}), \; (s + t) \in \operatorname{support}(\varpi^{\mathrm{c}})\} \; .$$

118    *(iii) Moreover, for any $x \in \mathbb{R}^d$, we have $\rho_{\mathrm{T}}^{\mathrm{c}}(x) = \int \varpi^{\mathrm{c}}(t)\rho(\phi_t(x))\mathbf{J}_{\phi_t}(x)\mathrm{d}t > 0$.*

119    Note that **H3** implies that $\sup_{t \in \mathbb{R}} |\varpi^{\mathrm{c}}(t)| < +\infty$. **H3** is automatically satisfied for example in the
120    case $\varpi^{\mathrm{c}} = \mathbb{1}_{[-T_1, T_2]}$ for $T_1, T_2 \geq 0$.

121   **Theorem S4.** *Assume H1, H2, H3. For any $x \in \mathbb{R}^d$ and $f : \mathbb{R}^d \to \mathbb{R}$ continuous and bounded,*

$$\lim_{h \downarrow 0} \left| \sum_{k \in \mathbb{Z}} w_{k,h}(x)f(\mathrm{T}_h^k(x)) - \int_{-\infty}^{\infty} w_t^{\mathrm{c}}(x)f(\phi_t(x))\mathrm{d}t \right| = 0 \; ,$$

122    *where $\{w_{k,h}\}_{k \in \mathbb{Z}}$ and $w_t^{\mathrm{c}}$ are defined in (S16) and (17) respectively, i.e. for $x \in \mathbb{R}^d$ and $t \in \mathbb{R}$,*

$$w_t^{\mathrm{c}}(x) = \varpi^{\mathrm{c}}(t)\rho(\phi_t(x))\mathbf{J}_{\phi_t}(x) \Big/ \int_{-\infty}^{\infty} \varpi^{\mathrm{c}}(s + t)\rho(\phi_s(x))\mathbf{J}_{\phi_s}(x)\mathrm{d}s \; . \tag{S20}$$

123    *Proof.* Let $f$ be a bounded continuous function, $x \in \mathbb{R}^d$. Setting

$$g_{k,h}(x) = \rho(\mathrm{T}_h^k(x))\varpi^{\mathrm{c}}(kh)\mathbf{J}_{\mathrm{T}_h^k}(x)f(\mathrm{T}_h^k(x))$$

$$h\Delta_{k,h}(x) = h \sum_{i \in \mathbb{Z}} \rho(\mathrm{T}_h^i(x))\varpi^{\mathrm{c}}((k + i)h)\mathbf{J}_{\mathrm{T}_h^i(x)} \; ,$$

124    we have that

$$\sum_{k \geq 0} \frac{hg_{k,h}(x)}{h\Delta_{k,h}(x)} = \int_0^{T_\varpi} \frac{1}{h\Delta_{\lfloor t/h \rfloor, h}(x)} g_{\lfloor t/h \rfloor, h}(x)\mathrm{d}t + \int_{T_\varpi}^{h\lfloor T_\varpi/h \rfloor + h} \frac{1}{h\Delta_{\lfloor t/h \rfloor, h}(x)} g_{\lfloor t/h \rfloor, h}(x)\mathrm{d}t \; ,$$

125    as $g_{k,h}(x) = 0$ when $k > \lfloor T_\varpi/h \rfloor$. Therefore, we can consider the following decomposition,

$$\left| \sum_{k \geq 0} \frac{\rho(\mathrm{T}_h^k(x))\varpi^{\mathrm{c}}(kh)\mathbf{J}_{\mathrm{T}_h^k}(x)f(\mathrm{T}_h^k(x))}{\sum_{i \in \mathbb{Z}} \rho(\mathrm{T}_h^i(x))\varpi^{\mathrm{c}}((k + i)h)\mathbf{J}_{\mathrm{T}_h^i(x)}} - \int_0^{T_\varpi} \frac{\varpi^{\mathrm{c}}(t)\rho(\phi_t(x))\mathbf{J}_{\phi_t}(x)f(\phi_t(x))\mathrm{d}t}{\int \varpi^{\mathrm{c}}(t + s)\rho(\phi_s(x))\mathbf{J}_{\phi_s}(x)\mathrm{d}s} \right| \leq A + B$$

126    with

$$A = \left| \int_0^{T_\varpi} \frac{1}{h\Delta_{\lfloor t/h \rfloor, h}(x)} \left\{ g_{\lfloor t/h \rfloor, h}(x) - \varpi^{\mathrm{c}}(t)\rho(\phi_t(x))\mathbf{J}_{\phi_t}(x)f(\phi_t(x)) \right\} \mathrm{d}t \right|$$

$$+ \left| \int_{T_\varpi}^{h\lfloor T_\varpi/h \rfloor + h} \frac{1}{h\Delta_{\lfloor t/h \rfloor, h}(x)} g_{\lfloor t/h \rfloor, h}(x)\mathrm{d}t \right| \; ,$$

127    and

$$B = \int_0^{T_\varpi} \left| \frac{\varpi^{\mathrm{c}}(t)\rho(\phi_t(x))\mathbf{J}_{\phi_t}(x)f(\phi_t(x))\mathrm{d}t}{h\Delta_{\lfloor t/h\rfloor,h}(x)} - \frac{\varpi^{\mathrm{c}}(t)\rho(\phi_t(x))\mathbf{J}_{\phi_t}(x)f(\phi_t(x))}{\int \varpi^{\mathrm{c}}(t+s)\rho(\phi_s(x))\mathbf{J}_{\phi_s}(x)\mathrm{d}s} \right| \mathrm{d}t \ ,$$

128    We bound those terms separately. First of all, under **H**3-(ii), for any $k$ such that $kh \in [0,T_\varpi]$, we have
129    $h\Delta_{k,h}(x) \geq hm^{-1}\Delta_{0,h}(x)$. Second, as $\lim_{h\downarrow 0} h\Delta_{0,h}(x) = \int_0^{T_\varpi}\rho(\phi_s(x))\mathbf{J}_{\phi_s}(x)\varpi^{\mathrm{c}}(s)\mathrm{d}s > 0$,
130    there exists some $\tilde{h} > 0$ and $c > 0$ such that for all $k \in \mathbb{Z}$, $h < \tilde{h}$ implies

$$\int_0^{T_\varpi} \varpi^{\mathrm{c}}(t)\rho(\phi_t(x))\mathbf{J}_{\phi_t}(x)\mathrm{d}t > c \ , \quad h\Delta_{k,h}(x) \geq hm^{-1}\Delta_{0,h}(x) > c \ . \tag{S21}$$

131    Then, for $h < \tilde{h}$,

$$A \leq c^{-1}\int_0^{T_\varpi} |g_{\lfloor t/h\rfloor,h}(x) - \varpi^{\mathrm{c}}(t)\rho(\phi_t(x))\mathbf{J}_{\phi_t}(x)f(\phi_t(x))|\mathrm{d}t$$

$$+ c^{-1}\int_{T_\varpi}^{h\lfloor T_\varpi/h\rfloor+h} \left|g_{\lfloor t/h\rfloor,h}(x)\right|\mathrm{d}t \ .$$

132    By **H**1 and **H**3, the function $t \to \varpi^{\mathrm{c}}(t)\rho(\phi_t(x))\mathbf{J}_{\phi_t}(x)f(\phi_t(x))$ is continuous on the compact
133    $[0,2T_\varpi]$ and thus is bounded. Therefore, for any $h \in \left(0,\bar{h}\right)$,

$$\sup_{t\in[0,2T_\varpi]} |\varpi^{\mathrm{c}}(t)\rho(\phi_t(x))\mathbf{J}_{\phi_t}(x)f(\phi_t(x))| \leq \sup_{t\in\mathbb{R}}|\varpi^{\mathrm{c}}| \sup_{x\in\mathbb{R}^d}|f(x)| \sup_{t\in[0,2T_\varpi]}|\rho(\phi_t(x))\mathbf{J}_{\phi_t}(x)| < \infty \ . \tag{S22}$$

134    Under **H**2, (S22) and Lemma S8 imply that

$$\sup_{t\in[0,h\lfloor T_\varpi/h\rfloor+h)} g_{\lfloor t/h\rfloor,h}(x)$$

$$\leq \sup_{t\in\mathbb{R}}|\varpi^{\mathrm{c}}(t)| \sup_{x\in\mathbb{R}^d}|f(x)| \sup_{t\in[0,h\lfloor T_\varpi/h\rfloor+h)}\rho(\mathrm{T}_h^{\lfloor t/h\rfloor}(x))\mathbf{J}_{\mathrm{T}_h^{\lfloor t/h\rfloor}}(x) < \infty \ ,$$

135    Then, $\lim_{h\downarrow 0}\int_{T_\varpi}^{h\lfloor T_\varpi/h\rfloor+h}\left|g_{\lfloor t/h\rfloor,h}(x)\right|\mathrm{d}t = 0$. Finally, Lemma S9 implies that $\lim_{h\downarrow 0}A = 0$.

136    Moreover, setting for $t \in [0,T_\varpi]$,

$$\Delta_{t,h}^B(x) \tag{S23}$$

$$= \int |\rho(\phi_{h\lfloor s/h\rfloor}(x))\varpi^{\mathrm{c}}(h(\lfloor s/h\rfloor+\lfloor t/h\rfloor))\mathbf{J}_{\phi_{h\lfloor s/h\rfloor}}(x) - \varpi^{\mathrm{c}}(s+t)\rho(\phi_s(x))\mathbf{J}_{\phi_s}(x)|\mathbb{1}_{\mathsf{A}_\varpi}(s,t)\mathrm{d}s$$

$$+ \int_{T_\varpi-h\lfloor t/h\rfloor}^{h(\lfloor T_\varpi/h\rfloor-\lfloor t/h\rfloor+1)} |\rho(\phi_{h\lfloor s/h\rfloor}(x))\varpi^{\mathrm{c}}(h(\lfloor s/h\rfloor+\lfloor t/h\rfloor))\mathbf{J}_{\phi_{h\lfloor s/h\rfloor}}(x)|\mathbb{1}_{\mathsf{A}_\varpi}(s,t)\mathrm{d}s \ ,$$

137    we have for $h < \tilde{h}$, by (S21) and **H**3-(ii),

$$B = \int_0^{T_\varpi} \left| \frac{\varpi^{\mathrm{c}}(t)\rho(\phi_t(x))\mathbf{J}_{\phi_t}(x)f(\phi_t(x))}{h\Delta_{\lfloor t/h\rfloor,h}(x)} - \frac{\varpi^{\mathrm{c}}(t)\rho(\phi_t(x))\mathbf{J}_{\phi_t}(x)f(\phi_t(x))}{\int \varpi^{\mathrm{c}}(s+t)\rho(\phi_s(x))\mathbf{J}_{\phi_s}(x)\mathrm{d}s} \right| \mathrm{d}t$$

$$\leq \int_0^{T_\varpi} \frac{\varpi^{\mathrm{c}}(t)\rho(\phi_t(x))\mathbf{J}_{\phi_t}(x)f(\phi_t(x))}{h\Delta_{\lfloor t/h\rfloor,h}(x)\int \varpi^{\mathrm{c}}(s+t)\rho(\phi_s(x))\mathbf{J}_{\phi_s}(x)\mathrm{d}s}\Delta_{t,h}^B(x)\mathrm{d}t$$

$$\leq mc^{-2}\int_0^{T_\varpi}\varpi^{\mathrm{c}}(t)\rho(\phi_t(x))\mathbf{J}_{\phi_t}(x)f(\phi_t(x))\Delta_{t,h}^B(x)\mathrm{d}t$$

$$\leq mc^{-2}\sup_{t\in\mathbb{R}}|\varpi^{\mathrm{c}}(t)|\sup_{x\in\mathbb{R}^d}|f(x)|\sup_{t\in[0,T_\varpi]}|\rho(\phi_s(x))\mathbf{J}_{\phi_s}(x)|\int_0^{T_\varpi}\Delta_{t,h}^B(x)\mathrm{d}t \ . \tag{S24}$$

138    By **H**1 and **H**3, the function $s \to \rho(\phi_s(x))\mathbf{J}_{\phi_s}(x)$ is continuous on the interval $[-T_\varpi,T_\varpi]$ and thus
139    is bounded. Therefore, for any $h \in \left(0,\bar{h}\right)$,

$$\sup_{(s,t)\in\mathsf{A}_\varpi}|\varpi^{\mathrm{c}}(h(\lfloor t/h\rfloor+\lfloor s/h\rfloor))\rho(\phi_{h\lfloor s/h\rfloor}(x))\mathbf{J}_{\phi_{h\lfloor s/h\rfloor}}(x)|$$

$$\leq \sup_{(s,t)\in\mathsf{A}_\varpi}|\varpi^{\mathrm{c}}(s+t)\rho(\phi_s(x))\mathbf{J}_{\phi_s}(x)| < T_\varpi\sup_{s\in\mathbb{R}}|\varpi^{\mathrm{c}}(s)|\sup_{s\in[-T_\varpi,T_\varpi]}|\rho(\phi_s(x))\mathbf{J}_{\phi_s}(x)| < \infty \ . \tag{S25}$$

140 This implies that

$$\lim_{h\downarrow 0}\int_{T_\varpi-h\lfloor t/h\rfloor}^{h(\lfloor T_\varpi/h\rfloor-\lfloor t/h\rfloor+1)}|\rho(\phi_{h\lfloor s/h\rfloor}(x))\varpi^c(h(\lfloor s/h\rfloor+\lfloor t/h\rfloor))\mathbf{J}_{\phi_{h\lfloor s/h\rfloor}(x)}|\mathrm{d}s=0\ .$$

Moreover, for any $t\in[0,T_\varpi]$, the function

$$s\mapsto|\varpi^c(h(\lfloor t/h\rfloor+\lfloor s/h\rfloor))\rho(\phi_{h\lfloor s/h\rfloor}(x))\mathbf{J}_{\phi_{h\lfloor s/h\rfloor}}(x)-\varpi^c(t+s)\rho(\phi_s(x))\mathbf{J}_{\phi_s}(x)|\mathbb{1}_{\mathrm{A}_\varpi}(s,t)$$

converges pointwise to $0$ for almost all $s\in\mathbb{R}$ when $h\downarrow 0$ using **H1**, **H3** and the continuity of $s\mapsto\phi_s(x)$. The Lebesgue dominated convergence theorem applies and by (S23), for all $t\in[0,T_\varpi]$,

$$\lim_{h\downarrow 0}\Delta_{t,h}^B(x)=0\ .$$

141 Moreover, using $h\Delta_{k,h}(x)=h\sum_{i\in\mathbb{Z}}\rho(\mathrm{T}_h^i(x))\varpi^c((k+i)h)\mathbf{J}_{\mathrm{T}_h^i(x)}$ and (S25),

$$\sup_{t\in[0,T_\varpi]}\sup_{h\in(0,\bar{h})}\Delta_{t,h}^B(x)<\infty\ .$$

142 The Lebesgue dominated convergence theorem and (S24) show that $\lim_{h\downarrow 0}B=0$ which concludes
143 the proof. $\qquad\square$

### B.1.1 Supporting Lemmas

145 For $f\in\mathrm{C}^1(\mathbb{R}^d,\mathbb{R}^d)$, define $\mathfrak{J}_f(x)$ the Jacobian matrix of $f$ evaluated at $x$ and the divergence
146 operator by $\mathrm{div}\,f(x)=\mathrm{tr}[\mathfrak{J}_f(x)]$.

**Lemma S5.** *Let $b$ be a $\mathrm{C}^1$ vector field in $\mathbb{R}^d$ and $(\phi_t)_{t\in\mathbb{R}}$ be the flow of the ODE (S17). For any $t\in\mathbb{R}$, the Jacobian of $\phi_t$ is given by*

$$\mathbf{J}_{\phi_t}(x)=\exp(\textstyle\int_0^t\mathrm{div}\,b(\phi_s(x))\mathrm{d}s)\ .$$

147 *Proof.* First, for $t\in\mathbb{R}$ and $x\in\mathbb{R}$, write $A(t,x)=\mathfrak{J}_{\phi_t}(x)$ the Jacobian matrix of $\phi_t$ evaluated
148 at $x$. By Jacobi's formula, $\mathrm{det}\,A(t,x)=\mathrm{tr}[\mathrm{adj}(A(t,x))\cdot\dot{A}(t,x)]$, where $\mathrm{tr}[M]$ denotes the trace
149 of a matrix $M$ and $\mathrm{adj}(M)$ its adjugate, i.e. the transpose of the cofactor matrix of $M$ such that
150 $\mathrm{adj}(M)M=\mathrm{det}(M)\,\mathrm{Id}$. Since for all $t$ and $x$, $\dot{A}(t,x)=\mathfrak{J}_{b\circ\phi_t}(x)=\mathfrak{J}_b(\phi_t(x))\cdot A(t,x)$, then

$$\dot{\mathbf{J}}_{\phi_t}(x)=\mathrm{tr}[\mathrm{adj}(A(t,x))\cdot\mathfrak{J}_b(\phi_t(x))\cdot A(t,x)]=\mathrm{tr}[\mathfrak{J}_b(\phi_t(x))]\mathbf{J}_{\phi_t}(x)\ .\qquad\text{(S26)}$$

151 Integrating this ODE yields $\mathbf{J}_{\phi_t}(x)=\exp(\int_0^t\mathrm{div}\,b(\phi_s(x))\mathrm{d}s)$. $\qquad\square$

152 **Lemma S6.** *Assume **H1**. Then, there exists $C>0$ such that for any $x\in\mathbb{R}^d,t\in\mathbb{R}$, $k\in\mathbb{Z}$, $h>0$,*

$$\|\phi_t(x)\|\leq Ce^{C|t|}(\|x\|+1)\ ,$$
$$\|\mathrm{T}_h^k(x)\|\leq Ce^{C|kh|}(\|x\|+1)\ .$$

153 This lemma follows from Gronwall's inequality and **H1**.

154 **Lemma S7.** *Assume **H1** and **H2**-(i). There exists $C>0$ such that for any $x\in\mathbb{R}^d,h\in(0,\bar{h})$,*

$$\|\mathrm{T}_h(x)-\phi_h(x)\|\leq C\{1+\|x\|\}\|h^{1+\delta}\ .\qquad\text{(S27)}$$

155 *Proof.* Under **H1** and **H2**-(i), we have

$$\|\mathrm{T}_h(x)-\phi_h(x)\|\leq\|x+hb(x)-\phi_h(x)\|+C_Fh^{1+\delta}(1+\|x\|)\ ,$$

156 and as $\phi_h(x)=x+\int_0^h b(\phi_s(x))\mathrm{d}s$,

$$\|x+hb(x)-\phi_h(x)\|=\|hb(x)-\textstyle\int_0^h b(\phi_s(x))\|\leq hL_b\sup_{s\in[0,h]}\|\phi_s(x)-x\|$$
$$\leq L_bh^2\{L_b\sup_{s\in[0,h]}\phi_s(x)+\|b(0)\|\}\ .\qquad\text{(S28)}$$

157 The proof is completed using Lemma S6. $\qquad\square$

**Lemma S8.** *Assume **H1** and **H2**-(i). There exists $C > 0$ such that for any $x \in \mathbb{R}^d, k \in \mathbb{N}, h \in (0, \bar{h})$, $kh \le T_\varpi$,*

$$\| T_h^k(x) - \phi_{kh}(x) \| \le Ce^{khC}(1 + \|x\|)h^\delta \ . \tag{S29}$$

*Proof.* Using Lemma S7, **H1** and **H2**-(i), there exist $C_1, C_2, C_3 > 0$ such that for any $x \in \mathbb{R}^d, k \in \mathbb{N}, h \in (0, \bar{h}), kh \le T_\varpi$,

$$\begin{aligned}
\| T_h^{k+1}(x) - \phi_{(k+1)h}(x) \| &\le \| T_h^{k+1}(x) - T_h \circ \phi_{kh}(x) \| + \| T_h \circ \phi_{kh}(x) - \phi_{(k+1)h}(x) \| \\
&\le (1 + hL_b)\| T_h^k(x) - \phi_{kh}(x) \| \\
&\qquad + h^{1+\delta}C_1\{2 + \| T_h^k(x) \| + \|\phi_{kh}(x)\|\} + \| T_h \circ \phi_{kh}(x) - \phi_{(k+1)h}(x) \| \\
&\le (1 + hL_b)\| T_h^k(x) - \phi_{kh}(x) \| + h^{1+\delta}2C_1C_2e^{C_2T_\varpi}\{1 + \|x\|\} + C_3\{1 + \|\phi_{kh}(x)\|\}h^{1+\delta} \\
&\le (1 + hL_b)\| T_h^k(x) - \phi_{kh}(x) \| \\
&\qquad + h^{1+\delta}2C_1C_2e^{C_2T_\varpi}\{1 + \|x\|\} + C_3\{1 + C_2(1 + \|x\|)\}h^{1+\delta}e^{C_2T_\varpi} \\
&\le (1 + hL_b)\| T_h^k(x) - \phi_{kh}(x) \| + A_T\{1 + \|x\|\}h^{1+\delta} \ ,
\end{aligned}$$

with $A_T = (2C_1C_2 + C_3(1 + C_2))e^{C_2T_\varpi}$. A straightforward induction yields

$$\| T_h^k(x) - \phi_{kh}(x) \| \le \frac{(1 + hL_b)^k}{L_b}A_T(1 + \|x\|)h^\delta \ .$$

$\square$

**Lemma S9.** *Assume **H1**, **H2**, **H3**. For any $x \in \mathbb{R}^d$, and $f : \mathbb{R}^d \to \mathbb{R}^d$ bounded and continuous,*

$$\lim_{h \downarrow 0} \int_0^{T_\varpi} \left| \varpi^c(h \lfloor t/h \rfloor)\rho(T_h^{\lfloor t/h \rfloor}(x))\mathbf{J}_{T_h^{\lfloor t/h \rfloor}}(x)f(T_h^{\lfloor t/h \rfloor}(x)) - \varpi^c(t)\rho(\phi_t(x))\mathbf{J}_{\phi_t}(x)f(\phi_t(x)) \right| \mathrm{d}t = 0 \ .$$

*Proof.* Let $x \in \mathbb{R}^d$. Consider the following decomposition, for any $h < \bar{h}$,

$$\begin{aligned}
&\int_0^{T_\varpi} \left| \varpi^c(h \lfloor t/h \rfloor)\rho(T_h^{\lfloor t/h \rfloor}(x))\mathbf{J}_{T_h^{\lfloor t/h \rfloor}}(x)f(T_h^{\lfloor t/h \rfloor}(x)) - \varpi^c(t)\rho(\phi_t(x))\mathbf{J}_{\phi_t}(x)f(\phi_t(x)) \right| \mathrm{d}t \\
&\le \frac{h}{T_\varpi}\sum_{k \in \mathbb{Z}} \varpi^c(kh)|\rho(T_h^k(x))\mathbf{J}_{T_h^k}(x)f(T_h^k(x)) - \rho(\phi_{kh}(x))\mathbf{J}_{\phi_{kh}}(x)f(\phi_{kh}(x))| \\
&+ \int_0^{T_\varpi} |\varpi^c(t)\rho(\phi_t(x))\mathbf{J}_{\phi_t}(x)f(\phi_t(x)) - \varpi^c(h \lfloor t/h \rfloor)\rho(\phi_{h\lfloor t/h \rfloor}(x))\mathbf{J}_{\phi_{h\lfloor t/h \rfloor}}(x)f(\phi_{h\lfloor t/h \rfloor}(x))|\mathrm{d}t \ .
\end{aligned}$$

The first term converges to 0 by Lemma S8 and **H2**-(ii) as $\varpi^c(kh) = 0$ for $kh > T_\varpi$. By **H1** and **H3**, the function $t \to \varpi^c(t)\rho(\phi_t(x))\mathbf{J}_{\phi_t}(x)f(\phi_t(x))$ is continuous on the compact $[0, T_\varpi]$ and thus is bounded. Therefore, for any $h \in (0, \bar{h})$,

$$\begin{aligned}
\sup_{t \in [0, T_\varpi]} &|\varpi^c(h \lfloor t/h \rfloor)\rho(\phi_{h\lfloor t/h \rfloor}(x))\mathbf{J}_{\phi_{h\lfloor t/h \rfloor}}(x)f(\phi_{h\lfloor t/h \rfloor}(x))| \\
&\le \sup_{t \in \mathbb{R}} |\varpi^c| \sup_{x \in \mathbb{R}^d} |f(x)| \sup_{t \in [0, T_\varpi]} |\rho(\phi_t(x))\mathbf{J}_{\phi_t}(x)| < \infty \ . \tag{S30}
\end{aligned}$$

Moreover, $t \mapsto \varpi^c(h \lfloor t/h \rfloor)\rho(\phi_{h\lfloor t/h \rfloor}(x))\mathbf{J}_{\phi_{h\lfloor t/h \rfloor}}(x)f(\phi_{h\lfloor t/h \rfloor}(x))$ converges pointwise when $h \downarrow 0$ to $t \to \varpi^c(t)\rho(\phi_t(x))\mathbf{J}_{\phi_t}(x)f(\phi_t(x))$ by continuity, using **H1** and **H3**. The Lebesgue dominated convergence theorem applies and the second term goes to 0 as $h \downarrow 0$. $\square$

## B.2 NEIS algorithm after [16]

Non Equilibrium Importance Sampling (NEIS) has been introduced in the pioneering work of [16]. NEIS relies on the flow of the ODE $\dot{x}_t = b(x_t)$ and the introduction of a set $\mathsf{O} \subset \mathbb{R}^d$. As in Appendix B, we assume **H1** holds and denote by $(\phi_t)_{t \in \mathbb{R}}$ the flow of this ODE.

Define for $x \in \mathsf{O}$, the exit times $\tau^+(x) \ge 0$ (resp. $\tau^-(x) \le 0$) satisfying

$$\tau^+(x) = \inf\{t \ge 0 : \phi_t(x) \notin \mathsf{O}\} \ , \ \tau^-(x) = \inf\{t \le 0 : \phi_t(x) \notin \mathsf{O}\} \ . \tag{S31}$$

The validity of NEIS relies on the following assumption.

178  **H4.** *The average time of an orbit in $\mathsf{O}$ is finite, i.e.*

$$Z_\tau = \int_{\mathsf{O}} (\tau^+(x) - \tau^-(x))\rho(x)\mathrm{d}x < \infty \ . \tag{S32}$$

179  Under **H4**, we can define the proposal distribution

$$\rho_{\mathrm{T}}(x) = Z_\tau^{-1} \int_{\mathsf{O}} \mathbb{1}_{[\tau^-(x),\tau^+(x)]}(t)\rho(\phi_t(x))\mathbf{J}_{\phi_t}(x)\mathrm{d}t \ . \tag{S33}$$

180  Under **H4**, [16, Equation (8)] derive the following estimator of $\rho(f)$, closely related to (16), in the
181  case $\varpi \equiv 1$, on the restricted set $\mathsf{O} \subset \mathbb{R}^d$ :

$$I_N^{\mathrm{NEIS}}(f) = \frac{1}{N} \sum_{i=1}^{N} \int_{\tau^-(X^i)}^{\tau^+(X^i)} w_t(X^i) f(\phi_t(X^i))\mathrm{d}t \tag{S34}$$

$$w_t(x) = \frac{\rho(\phi_t(x))\mathbf{J}_{\phi_t(x)}}{\int_{\tau^-(x)}^{\tau^+(x)} \rho(\phi_t(x))\mathbf{J}_{\phi_t(x)}\mathrm{d}t} \ . \tag{S35}$$

182  Note that in practice, in order for **H4** to be verified, one typically requires that $\mathsf{O}$ be bounded, as
183  discussed in [16].

184  Following [16], consider a $d$-dimensional system with position $q \in \mathbb{R}^d$, momentum $p \in \mathbb{R}^d$ and
185  Hamiltonian $H(p,q) = (1/2)\|p\|^2 + U(q)$ where $U(q)$ is a potential assumed to be bounded from
186  below. Denote by $V(E)$ the volume of the phase-space below some threshold energy $E$,

$$V(E) = \int \mathbb{1}_{\{H(p,q)\leq E\}}\mathrm{d}p\mathrm{d}q \ . \tag{S36}$$

187  To calculate (S36), we set $x = (p,q)$, define $\mathsf{O} = \{x; H(x) \leq E_{\max}\}$ for some $E_{\max} < \infty$, and use
188  the dissipative Langevin dynamics with $b(x) = (p, -\nabla U(q) - \gamma p)$, *i.e.*

$$\dot{q} = p \ , \quad \dot{p} = -\nabla U(q) - \gamma p \ ,$$

189  for some friction coefficient $\gamma > 0$. With this choice, $\mathbf{J}_{\phi_t}(x) = \mathrm{e}^{-d\gamma t}$. Taking $\rho$ to be the
190  uniform distribution on the (bounded) set $\mathsf{O}$, write the estimator for $E \leq E_{\max}$, $V(E)/V(E_{\max}) = $
191  $\int \mathbb{1}_{\{H(p,q)\leq E\}}\rho(p,q)\mathrm{d}p\mathrm{d}q$, where $\rho(p,q) = \mathbb{1}_{\mathsf{O}}(p,q)/V(E_{\max})$, we get

$$\begin{aligned}
V(E)/V(E_{\max}) &= \frac{1}{N} \sum_{i=1}^{N} \frac{\int_{\tau^-(X^i)}^{\tau^+(X^i)} \mathbf{J}_{\phi_t(X^i)} \mathbb{1}_{\{H(\phi_t(X^i))\leq E\}}\mathrm{d}t}{\int_{\tau^-(X^i)}^{\tau^+(X^i)} \mathbf{J}_{\phi_t(X^i)}\mathrm{d}t} \\
&= \frac{1}{N} \sum_{i=1}^{N} \frac{\int_{\tau^E(X^i)}^{\tau^+(X^i)} \mathbf{J}_{\phi_t(X^i)}\mathrm{d}t}{\int_{\tau^-(X^i)}^{\tau^+(X^i)} \mathbf{J}_{\phi_t(X^i)}\mathrm{d}t} = \frac{1}{N} \sum_{i=1}^{N} \mathrm{e}^{-d\gamma(\tau^E(X^i)-\tau^-(X^i))} \ ,
\end{aligned} \tag{S37}$$

192  where $\tau^E(x)$ denotes the (possibly infinite) time for a trajectory initiated at $x = (p,q)$ to reach the
193  energy $E \leq E_{\max}$.

194  Finally, to estimate the normalizing constant, [16] discretize the energy levels $\{E_0, \ldots, E_P\}$ and
195  write their estimator as

$$\widehat{Z}_{X^{1:N}}^{\mathrm{NEIS}} = \frac{1}{N} \sum_{i=1}^{N} \sum_{\ell=1}^{P} \mathrm{e}^{-d\gamma(\tau_\ell^E(X^i)-\tau^-(X^i))}(E_\ell - E_{\ell-1}) \ , \tag{S38}$$

196  using an approximation of the identity

$$Z = \int_{\mathsf{O}} \int_0^\infty \mathbb{1}_{\{\mathrm{L}(x)>L\}}\rho(x)\mathrm{d}L\mathrm{d}x = \int_0^\infty \mathbb{P}_{X\sim\rho}(\mathrm{L}(X) > L)\mathrm{d}L \ ,$$

197  which is at the core of nested sampling [5].

 **B.3** NEO **with exit times**

199 Consider $O \subset \mathbb{R}^d$ and let T be a $C^1$-diffeomorphism on $\mathbb{R}^d$. We introduce here an estimator based
200 on the forward and backward orbits in O associated with T. Define the exit times $\tau^+ : \mathbb{R}^d \to \mathbb{N}$ and
201 $\tau^- : \mathbb{R}^d \to \mathbb{N}_-$, given, for all $x \in \mathbb{R}^d$, by

$$\tau^+(x) = \inf\{k \geq 1 \, : \, T^k(x) \notin O\} \, , \tag{S39}$$

$$\tau^-(x) = \sup\{k \leq -1 \, : \, T^k(x) \notin O\} \, , \tag{S40}$$

202 with the convention $\inf \emptyset = +\infty$ and $\sup \emptyset = -\infty$, and set

$$I = \{(x, k) \in O \times \mathbb{Z} \, : \, k \in [\tau^-(x) + 1 : \tau^+(x) - 1]\} \, . \tag{S41}$$

203 For any $k \in \mathbb{Z}$, define $\rho_k : \mathbb{R}^d \to \mathbb{R}_+$ by

$$\rho_k(x) = \rho(T^{-k}(x)) \mathbf{J}_{T^{-k}}(x) \mathbb{1}_I(x, -k) \, . \tag{S42}$$

204 The density $\rho_k$ is the push-forward of $\mathbb{1}_I(x, k)\rho(x)$ by $T^k$, *i.e.* for any $k \in \mathbb{Z}$ and any bounded
205 function $g : \mathbb{R}^d \to \mathbb{R}$,

$$\int_O g(y)\rho_k(y)\mathrm{d}y = \int_O g(T^k(x)) \mathbb{1}_I(x, k)\rho(x)\mathrm{d}x \, . \tag{S43}$$

206 Consider the following assumption:

207 **H5.** *The nonnegative sequence $(\varpi_k)_{k \in \mathbb{Z}}$ satisfies $\varpi_0 > 0$ and*

$$Z_T^\varpi = \int_O \sum_{k \in \mathbb{Z}} \varpi_k \rho_k(x)\mathrm{d}x = \int_O \sum_{k \in \mathbb{Z}} \varpi_k \rho(T^k(x)) \mathbf{J}_{T^k}(x) \mathbb{1}_I(x, k)\mathrm{d}x < \infty \, . \tag{S44}$$

208 Consider the pdf

$$\rho_T(x) = \frac{1}{Z_T^\varpi} \sum_{k \in \mathbb{Z}} \varpi_k \rho_k(x) \, , \tag{S45}$$

209 where $Z_T^\varpi$ is the normalizing constant. This is a *non-equilibrium* distribution, since $\rho_T$ is not
210 invariant by T in general. Using $\rho_T$ as an importance distribution to obtain an unbiased estimator of
211 $\int f(x)\rho(x)\mathrm{d}x$ is feasible since as $\varpi_0 > 0$, $\sup_{x \in O} \rho(x)/\rho_T(x) \leq Z_T/\varpi_0 < \infty$, hence

$$\int_O f(x)\rho(x)\mathrm{d}x = \int_O \left(f(x)\frac{\rho(x)}{\rho_T(x)}\right)\rho_T(x)\mathrm{d}x \, .$$

212 From (S43), the right hand side can be computed using the following key result.

213 **Theorem S10.** *For any $f : \mathbb{R}^d \to \mathbb{R}$ measurable bounded function, we have*

$$\int_O f(x)\rho(x)\mathrm{d}x = \int_O \sum_{k \in \mathbb{Z}} f(T^k(x))w_k(x)\rho(x)\mathrm{d}x \, , \tag{S46}$$

214 *where, for any $x \in \mathbb{R}^d$ and $k \in \mathbb{Z}$,*

$$w_k(x) = \varpi_k \rho_{-k}(x) \Big/ \sum_{j \in \mathbb{Z}} \varpi_{j+k} \rho_j(x) \, . \tag{S47}$$

215 *Proof.* Let $f : \mathbb{R}^d \to \mathbb{R}$ be a measurable bounded function. By (S43), writing $g \leftarrow f\rho/\rho_T$,

$$\int_O f(x)\rho(x)\mathrm{d}x = \int_O \left(f(x)\frac{\rho(x)}{\rho_T(x)}\right)\rho_T(x)\mathrm{d}x$$

$$= \int_O \sum_{k \in \mathbb{Z}} \left(f(T^k(x))\frac{\varpi_k \rho(T^k(x)) \mathbb{1}_I(x, k)}{Z_T^\varpi \rho_T(T^k(x))}\right)\rho(x)\mathrm{d}x \, .$$

216 We now need to prove:

$$\frac{\varpi_k \rho(T^k(x)) \mathbb{1}_I(x, k)}{Z_T^\varpi \rho_T(T^k(x))} = \frac{\varpi_k \rho(T^k(x)) \mathbb{1}_I(x, k)}{\mathbb{1}_I(x, k) \sum_{i \in \mathbb{Z}} \varpi_i \rho_i(T^k(x))} = \frac{\varpi_k \rho_{-k}(x)}{\sum_{j \in \mathbb{Z}} \varpi_{j+k} \rho_j(x)} = w_k(x) \, ,$$

with the convention $0/0 = 0$. We thus need to show that for any $x \in \mathsf{O}$, $k \in \mathbb{Z}$,

$$\mathbb{1}_{\mathrm{I}}(x,k) \sum_{i \in \mathbb{Z}} \varpi_i \rho_i(\mathrm{T}^k(x)) = \frac{\mathbb{1}_{\mathrm{I}}(x,k)}{\mathbf{J}_{\mathrm{T}^k}(x)} \sum_{j \in \mathbb{Z}} \varpi_{j+k} \rho_j(x) \ .$$

Using the identity $\mathbf{J}_{\mathrm{T}^{-i+k}}(x) = \mathbf{J}_{\mathrm{T}^{-i}}(\mathrm{T}^k(x)) \mathbf{J}_{\mathrm{T}^k}(x)$, we obtain

$$
\begin{aligned}
\mathbb{1}_{\mathrm{I}}(x,k) \sum_{i \in \mathbb{Z}} \varpi_i \rho_i(\mathrm{T}^k(x)) &= \sum_{i \in \mathbb{Z}} \mathbb{1}_{\mathrm{I}}(x,k) \varpi_i \rho(\mathrm{T}^{-i}(\mathrm{T}^k(x))) \mathbf{J}_{\mathrm{T}^{-i}}(\mathrm{T}^k(x)) \mathbb{1}_{\mathrm{I}}(\mathrm{T}^k(x), -i) \\
&= \frac{1}{\mathbf{J}_{\mathrm{T}^k}(x)} \sum_{i \in \mathbb{Z}} \mathbb{1}_{\mathrm{I}}(x,k) \varpi_i \rho(\mathrm{T}^{-i+k}(x)) \mathbf{J}_{\mathrm{T}^{-i+k}}(x) \mathbb{1}_{\mathrm{I}}(\mathrm{T}^k(x), -i) \\
&= \frac{1}{\mathbf{J}_{\mathrm{T}^k}(x)} \sum_{j \in \mathbb{Z}} \varpi_{j+k} \rho(\mathrm{T}^{-j}(x)) \mathbf{J}_{\mathrm{T}^{-j}}(x) \mathbb{1}_{\mathrm{I}}(\mathrm{T}^k(x), -j-k) \mathbb{1}_{\mathrm{I}}(x,k)
\end{aligned}
$$

Note that if $(x,k) \in \mathrm{I}$, we have $(x,-j) \in \mathrm{I}$ if and only if $(\mathrm{T}^k(x), -j-k) \in \mathrm{I}$ by definition of I (S41). The proof is concluded by noting that:

$$\mathbb{1}_{\mathrm{I}}(\mathrm{T}^k(x)), -j-k) \mathbb{1}_{\mathrm{I}}(x,k) = \mathbb{1}_{\mathrm{I}}(x,-j) \mathbb{1}_{\mathrm{I}}(x,k) \ .$$

$\square$

## C  Iterated SIR

Let us recall the principle of the Sampling Importance Resampling method (SIR; Rubin [17], Smith & Gelfand [18]) whose goal is to approximately sample from the target distribution $\pi$ using samples drawn from a proposal distribution $\rho$.

In SIR, a $N$-i.i.d. sample $X^{1:N}$ is first generated from the proposal distribution $\rho$. A sample $X^*$ is approximately drawn from the target $\pi$ by choosing randomly a value in $X^{1:N}$ with probabilities proportional to the importance weights $\{\mathrm{L}(X^i)\}_{i=1}^N$, where $\mathrm{L}(x) = \pi(x)/\rho(x)$. Note that the importance weights are required to be known only up to a constant factor.

For SIR, as $N \to \infty$, the sample $X^*$ is *asymptotically* distributed according to $\pi$; see [18].

A subsequent algorithm is the *iterated SIR* (i-SIR) [2]. Here, $N$ is not necessarily large ($N \geq 2$), the whole process of sampling a set of proposals, computing the importance weights, and picking a candidate, is iterated. At the $n$-th step of i-SIR, the active set of $N$ proposals $X_n^{1:N}$ and the index $I_n \in [N]$ of the conditioning proposal are kept. First i-SIR updates the active set by setting $X_{n+1}^{I_n} = X_n^{I_n}$ (keep the conditioning proposal) and then draw independently $X_{n+1}^{1:N \setminus \{I_n\}}$ from $\rho$. Then it selects the next proposal index $I_{n+1} \in [N]$ by sampling with probability proportional to $\{\tilde{w}(X_{n+1}^i)\}_{i=1}^N$. As shown in [2], this algorithm defines a partially collapsed Gibbs sampler (PCG) of the augmented distribution

$$\bar{\pi}(x^{1:N}, i) = \frac{1}{N} \pi(x^i) \prod_{j \neq i} \rho(x^j) = \frac{1}{N} \tilde{w}(x^i) \prod_{j=1}^N \rho(x^j) \ .$$

The PCG sampler can be shown to be ergodic provided that $\rho$ and $\pi$ are continuous and $\rho$ is positive on the support of $\pi$. If in addition the importance weights are bounded, the Gibbs sampler can be shown to be uniformly geometrically ergodic [14, 3]. It follows that the distribution of the conditioning proposal $X_n^* = X_n^{I_n}$ converges to $\pi$ as the iteration index $n$ goes to infinity. Indeed, for any integrable function $f$ on $\mathbb{R}^d$, with $(X_{1:N}, I) \sim \bar{\pi}$,

$$\mathbb{E}[f(X^I)] = \int \sum_{i=1}^N f(x^i) \bar{\pi}(x^{1:N}, i) \mathrm{d}x^{1:N} = N^{-1} \sum_{i=1}^N \int f(x^i) \pi(x^i) \mathrm{d}x_i = \int f(x) \pi(x) \mathrm{d}x \ .$$

When the state space dimension $d$ increases, designing a proposal distribution $\rho$ guaranteeing proper mixing properties becomes more and more difficult. A way to circumvent this problem is to use dependent proposals, allowing in particular *local moves* around the conditioning orbit. To implement

this idea, for each $i \in [N]$, we define a proposal transition, $r_i(x^i; x^{1:N \setminus \{i\}})$ which defines the the conditional distribution of $X^{1:N \setminus \{i\}}$ given $X^i = x^i$. The key property validating i-SIR with dependent proposals is that all one-dimensional marginal distributions are equal to $\rho$, which requires that for each $i, j \in [N]$,

$$\rho(x^i) r_i(x^i; x^{1:N \setminus \{i\}}) = \rho(x^j) r_j(x^j; x^{1:N \setminus \{j\}}) \tag{S48}$$

The (unconditional) joint distribution of the particles is therefore defined as

$$\rho_N\left(x^{1:N}\right) = \rho(x^1) r_1(x^1; x^{1:N \setminus \{1\}}) . \tag{S49}$$

The resulting modification of the i-SIR algorithm is straightforward: $X^{1:N \setminus \{I_n\}}$ is sampled jointly from the conditional distribution $r_{I_n}(X_n^{I_n}, \cdot)$ rather than independently from $\rho$.

# D   Additional Experiments

## D.1   Normalizing constant estimation

We consider here the problem of the estimation of the normalizing constant of Cauchy mixtures. The Cauchy distribution with scale $\sigma$ has a pdf defined by $\mathrm{Cauchy}(x; \mu, \sigma) = [\pi \sigma (1 + \{(x - \mu)/\sigma\}^2]^{-1}$. The target distribution is a product of mixtures of two Cauchy distributions,

$$\pi(x) = \prod_{i=1}^n \frac{1}{2} \left[\mathrm{Cauchy}\left(x_i; \mu, \sigma\right) + \mathrm{Cauchy}\left(x_i; -\mu, \sigma\right)\right], \quad \mu = 5, \sigma = 1 .$$

NEO-IS is compared with IS estimator using the same proposal $\rho$. We also compare NEO-IS to Neural IS [15] with a Cauchy as base distribution.

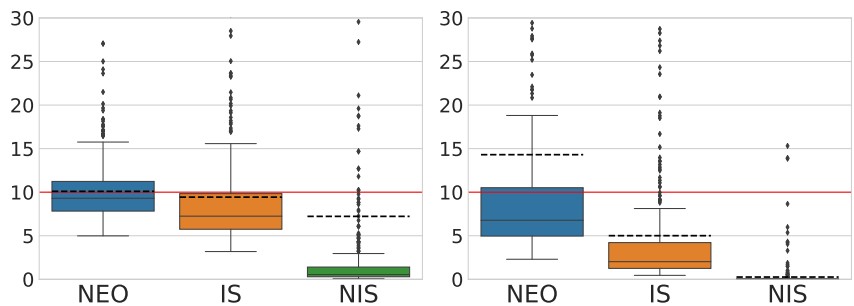

Figure S1: Boxplots of 500 independent estimations of the normalizing constant of the Cauchy mixture in dimension $d = 10, 15$ (top, bottom). The true value is given by the red line. The figure displays the median (solid lines), the interquartile range, and the mean (dashed lines) over the 500 runs

Finally, we compare NEO-IS with NEIS[1]. We consider here `MG25` in dimension 5 and 10, where all the covariances of the Gaussian distributions are diagonal and equal to $0.005\,\mathrm{Id}$. NEIS and NEO-IS are run for the same computational time. We add an IS scheme as a baseline for comparison. All algorithms (NEO-IS, NEIS, IS) are run for 7.20s and 11.30s wall clock time respectively for $d = 5$ and $d = 10$. For NEO-IS, we use a conformal transform with $h = 0.1$, $K = 10$ and $\gamma = 1$. For NEIS, we choose $\gamma = 1$ and consider a stepsize $h = 10^{-4}$ corresponding to an optimal trade-off between the discretization bias inherent to NEISand its computational budget. We can observe that NEO-IS always outperforms NEIS, which suffers from a non-negligeable bias if the stepsize $h$ is not chosen small enough.

## D.2   Gibbs inpainting

We display here additional results for the Gibbs inpainting experiment presented in Section 5. We emphasize that the starting images are chosen at random in the test set.

---

[1]The code from [16] we run is available at `https://gitlab.com/rotskoff/trajectory_estimators/-/tree/master`.

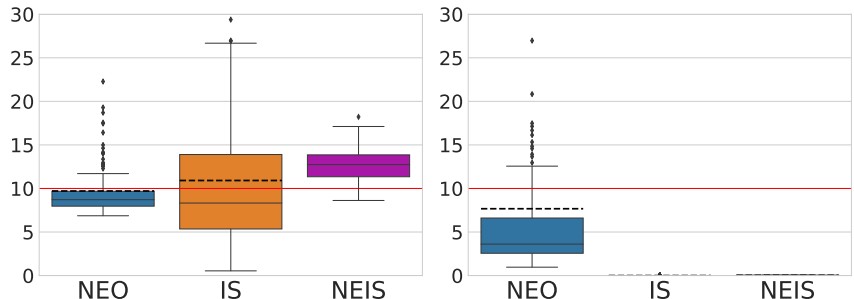

Figure S2: NEO v. NEIS. 25 GM with $\sigma^2 = 0.005$, $d = 5$. 500 runs each.

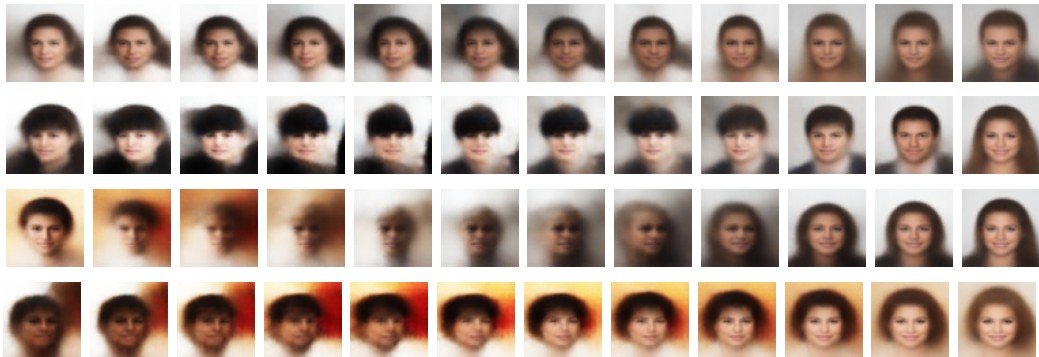

Figure S3: Forward orbits of NEO-MCMC.

## E  NEO **and VAEs**

Denote by $p_\theta(x, z)$ the joint distribution of the observation $z \in \mathbb{R}^p$ and the latent variable $x \in \mathbb{R}^d$. The marginal likelihood is given, for $z \in \mathbb{R}^p$ by $p_\theta(z) = \int p_\theta(x, z)\mathrm{d}x$. Given a training set $\mathcal{D} = \{z_i\}_{i=1}^M$, the objective is to estimate $\theta$ by maximizing the likelihood, *i.e.* maximizing $\log p_\theta(\mathcal{D}) = \sum_{i=1}^M \log p_\theta(z_i)$. We show two experiments in the following, first the evaluation of independently trained VAEs, and then the derivation and learning of a VAE based on NEO, and NEO-VAE.

### E.1  Log-likelihood estimation

We present here first the evaluation of the log-likelihood of a trained VAE on the dynamically binarized MNIST dataset. The models we compare share the same architecture: the inference network $q_\phi$ is given by a convolutional network with 2 convolutional layers and one linear layer, which outputs the parameters $\mu_\phi(x), \sigma_\phi(x) \in \mathbb{R}^d$ of a factorized Gaussian distribution, while the generative model $p_\theta(\cdot|z)$ is given by another symmetrical convolutional network $g_\theta$. This outputs the parameters for the factorized Bernoulli distribution (for MNIST dataset), that is

$$p_\theta(z|x) = \prod_{i=1}^N \mathrm{Ber}\left(z^{(i)}|\left(g_\theta(x)\right)^{(i)}\right).$$

We here follow the experimental setting of [20]. Given a test set $\mathcal{T} = \{z_i\}_{i=1}^{M_\mathcal{T}}$, we estimate $\sum_{i=1}^{M_\mathcal{T}} \log p_{\theta^*}(z_i)$. We also estimate similarly the log-likelihood of an Importance Weighted Auto Encoder (IWAE) [4]. Following [20], we compare IS, AIS, and NEO-IS. As previously, AIS, IS, and NEO-IS are given a similar computational budget, choosing here $K = 12$, $N = 5 \cdot 10^3$. For NEO, we choose $\gamma = 1.$ and $h = 0.2$. Similarly, the stepsize of HMC transitions in AIS is $h = 0.1$ in order to achieve an acceptance ratio of around $0.6$ in the HMC transitions. We report in Table 1 the log-likelihood computed on the test set for VAE, IWAE with latent dimension in $\{16, 32\}$. For

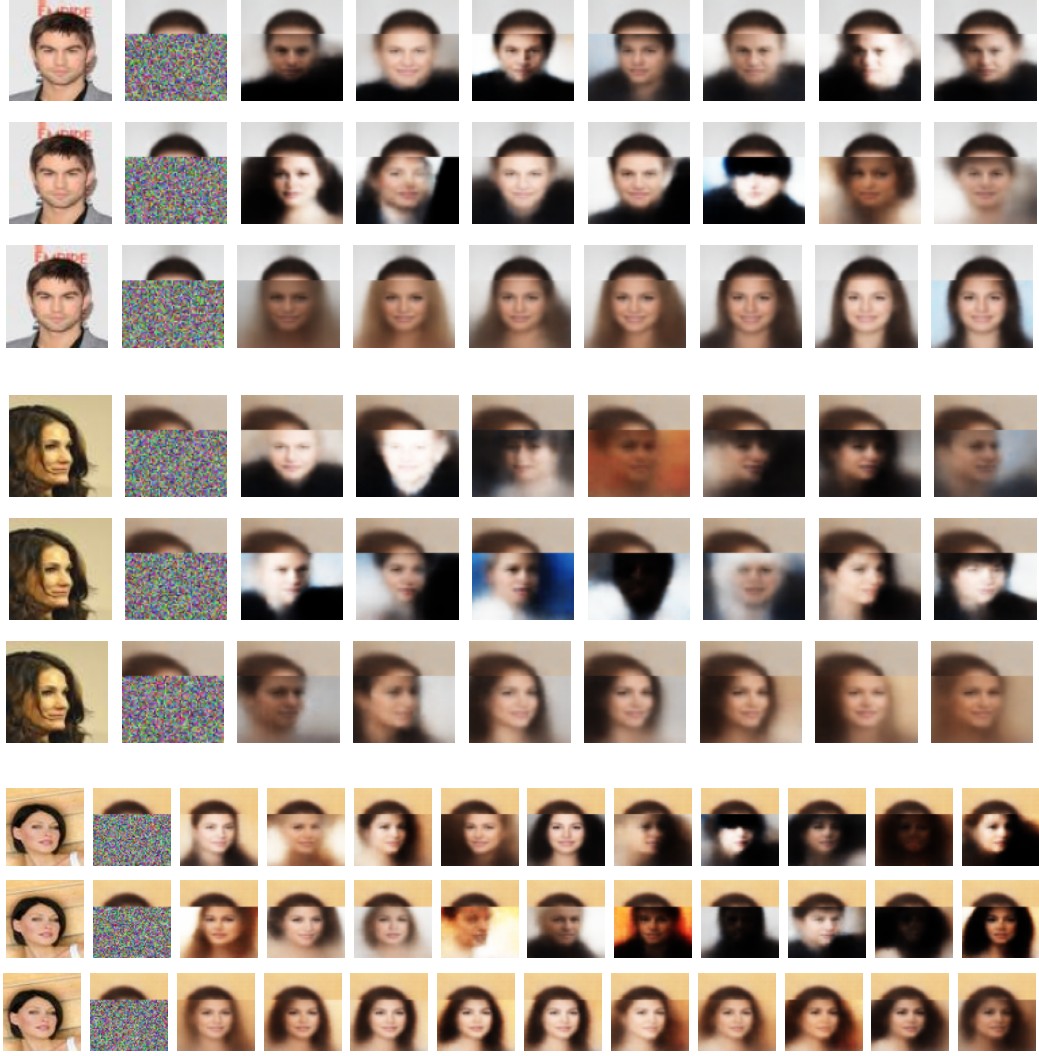

Figure S4: Additional examples for the Gibbs inpainting task for CelebA dataset. From top to bottom: i-SIR, HMC and NEO-MCMC: From left to right, original image, blurred image to reconstruct, and output every 5 iterations of the Markov chain.

| Model | VAE, $d = 32$ | VAE, $d = 16$ | IWAE, $d = 32$ | IWAE, $d = 16$ |
|-------|---------------|---------------|----------------|----------------|
| IS | -90.17 | -90.44 | -88.76 | -90.13 |
| AIS | -89.67 | -89.97 | -88.30 | -89.61 |
| NEO-IS | -88.81 | -89.17 | -87.46 | -88.99 |

Table 1: Evaluation of the log-likelihood (normalizing constant) of different Variational Auto Encoders.

the same computational budget, NEO-IS yields consistently better values for the estimation of the log-likelihood of the VAE.

## E.2 Definition of a NEO-VAE

Variational inference (VI) provides us with a tool to simultaneously approximate the intractable posterior $p_\theta(x|z)$ and maximize the marginal likelihood $p_\theta(\mathcal{D})$ in the parameter $\theta$. This is achieved by introducing a parametric family $\{q_\phi(x|z), \phi \in \Phi\}$ to approximate the posterior $p_\theta(x|z)$ and maximizing the Evidence Lower Bound (ELBO) (see [12]) $\mathcal{L}_{\mathrm{ELBO}}(\mathcal{D}, \theta, \phi) = \sum_{i=1}^{M} \mathcal{L}_{\mathrm{ELBO}}(z_i, \theta, \phi)$

where

$$\mathcal{L}_{\text{ELBO}}(z, \theta, \phi) = \int \log \left( \frac{p_\theta(x, z)}{q_\phi(x \mid z)} \right) q_\phi(x \mid z) \mathrm{d}x \tag{S50}$$

$$= \log p_\theta(z) - \text{KL}(q_\phi(\cdot \mid z) \| p_\theta(\cdot \mid z)) ,$$

and KL is the Kullback–Leibler divergence. In the sequel, we set $\rho(x) = q_\phi(x \mid z)$ and $\text{L}(x) = p_\theta(x, z)/q_\phi(x \mid z)$. In such a case, $\pi(x) = \rho(x)\text{L}(x)/\text{Z} = p_\theta(x \mid z)$ and $\text{Z} = p_\theta(z)$ (in these notations, the dependence in the observation $z$ is implicit).

We follow the the auxiliary variational inference framework (AVI) provided by [1]. We consider a joint distribution $\bar{p}_\theta(x, u, z)$ which is such that $p_\theta(z) = \int p_\theta(x, u, z) \mathrm{d}x \mathrm{d}u$ where $u \in \mathsf{U}$ is an auxiliary variable (the auxiliary variable can both have discrete and continuous components; when $u$ has discrete components the integrals should be replaced by a sum). Then as the usual VI approach, we consider a parametric family $\{\bar{q}_\phi(x, u|z), \phi \in \Phi\}$. Introducing auxiliary variables loses the tractability of (S50) but they allow for their own ELBO as suggested in [1, 13] by minimizing

$$\text{KL}(\bar{q}_\phi(\cdot \mid z) \| \bar{p}_\theta(\cdot \mid z)) = \int \bar{q}_\phi(x, u|z) \log \left( \frac{\bar{p}_\theta(x, u, z)}{\bar{q}_\phi(x, u|z)} \right) \mathrm{d}x \mathrm{d}u . \tag{S51}$$

The auxiliary variable $u$ is naturally associated with the extended target $\bar{p}$ defined similar to Remark 2,

$$\bar{p}_N([x, x^{1:N \setminus \{i\}}], i) = \check{\pi}(x^{1:N}, i) = \frac{\widehat{\text{Z}}_x^\varpi}{N \text{Z}} \rho_N(x^{1:N}) \tag{S52}$$

with $(x, u) = ([x, x^{1:N \setminus \{i\}}], i)$, a shorthand notation for a $N$-tuple $x^{1:N}$ with $x^i = x$, and, with $r_i$ defined in (15),

$$\rho_N(x^{1:N}) = \rho(x^1) r_1(x^1, x^{2:N}) = \rho(x^j) r_j(x^j, x^{1:N \setminus \{j\}}) , \quad j \in \{1, \ldots, N\} , \tag{S53}$$

generally for Markov transitions $\{r_j\}_{j \in [N]}$. We might write simply in the following

$$\rho_N(x^{1:N}) = \prod_{i=1}^N \rho(x^i) .$$

An extended proposal playing the role of $\bar{q}_\phi(x, u|z)$ is derived from the NEO-MCMC sampler, i.e.

$$\bar{q}_N([x, x^{1:N \setminus \{i\}}], i) = \frac{\widehat{\text{Z}}_x^\varpi}{N \widehat{\text{Z}}_{x^{1:N}}^\varpi} \rho_N(x^{1:N}) . \tag{S54}$$

where $\widehat{\text{Z}}_{x^{1:N}}^\varpi$ is the NEO estimator (4) of the normalizing constant. Note that, by construction,

$$\sum_{i=1}^N \bar{q}_N(x^{1:N}, i) = \rho_N(x^{1:N}) \tag{S55}$$

showing that this joint proposal can be sampled by drawing the proposals $x^{1:N} \sim \rho_N$, then sampling the path index $i \in [N]$ with probability proportional to $(\widehat{\text{Z}}_{x^i}^\varpi)_{i=1}^N$ (with $\widehat{\text{Z}}_x^\varpi$ defined in (4)). The ratio of (S52) over (S54) is

$$\bar{p}_N(x^{1:N}, i)/\bar{q}_N(x^{1:N}, i) = \widehat{\text{Z}}_{x^{1:N}}^\varpi / \text{Z} . \tag{S56}$$

Thus, we write the augmented ELBO (S51)

$$\mathcal{L}_{\text{NEO}} = \int \rho_N(x^{1:N}) \log \widehat{\text{Z}}_{x^{1:N}}^\varpi \mathrm{d}x^{1:N} = \log \text{Z} - \text{KL}(\bar{q}_N | \bar{p}_N) , \tag{S57}$$

where we have used (S55) and that the ratio $\bar{p}_N(x^{1:N}, i)/\bar{q}_N(x^{1:N}, i)$ does not depend on the path index $i$. When $\varpi_k = \delta_{k,0}$, where $\delta_{i,j}$ is the Kronecker symbol, and $\rho_N(x^{1:N}) = \prod_{j=1}^N \rho(x^j)$, we exactly retrieve the Importance Weighted AutoEncoder (IWAE); see e.g. [4] and in particular the interpretation in [6].

Choosing the conformal Hamiltonian introduced in Section 2 allows for a family of invertible flows that depends on the parameter $\theta$ which itself is directly linked to the target distribution. Table 2 displays the estimated NLL of all models provided by IS and the NEO method. It is interesting to note here again that NEO improves the training of the VAE when the dimension of the latent space is small to moderate.

Table 2: Negative Log Likelihood estimates for VAE models for different latent space dimensions.

| model | $d = 4$ | | $d = 8$ | | $d = 16$ | | $d = 50$ | |
|---|---|---|---|---|---|---|---|---|
| | IS | NEO | IS | NEO | IS | NEO | IS | NEO |
| VAE | 115.01 | 113.49 | 97.96 | 97.64 | 90.52 | 90.42 | 88.22 | 88.36 |
| IWAE, $N = 5$ | 113.33 | 111.83 | 97.19 | 96.61 | 89.34 | 89.05 | 87.49 | 87.27 |
| IWAE, $N = 30$ | 111.92 | 110.36 | 96.81 | 95.94 | 88.99 | 88.64 | 86.97 | 86.93 |
| NEO VAE, $K = 3$ | 109.14 | 107.47 | 94.50 | 94.26 | 89.03 | 88.92 | 88.14 | 88.16 |
| NEO VAE, $K = 10$ | 110.02 | 107.90 | 94.63 | 94.22 | 89.71 | 88.68 | 88.25 | 86.95 |