# OpenReview forum: "NEO: Non Equilibrium Sampling on the Orbits of a Deterministic Transform"
_NeurIPS.cc/2021/Conference — NeurIPS 2021 Poster_

### Official Review · Reviewer_ckCN · 2021-07-16

**Rating:** 6
**Confidence:** 3

**Summary:**

This paper proposes NEO which extends Non-Equilibrium Importance Sampling (NEIS) to use discrete orbit and extra weights.
The probability distribution and sampling process is derived based on the Jacobian determinant.
This method could be applied to estimate the partition function and a biased estimation called self-normalized IS is proposed based on estimated partition function.
This paper also derive upper bound for the bias and MSE.
A damped Hamiltonian system is adopted to equip the original system with extra momentum and provide orbit trajectory for NEO.
NEO is also combined to Sampling Importance Resampling to derive the NEO-MCMC procedure.
The authors present reasonable experiments to evaluate their methods.

**Limitations And Societal Impact:**

This article clearly describes the assumptions needed for each theoretical result.

**Main Review:**

Compared with NEIS, NEO not only adopt discrete orbit but also introduce an extra weight which could be of practical interest.
The paper itself is mostly clear, however certain proof in supplement is hard to follow.

Questions:
1. It seems the kernel in eq.15 doesn't satisfy condition in line 225-226. Is it just a typo?
2. Is there any efficiency guarantee for general NEO-IS estimator defined in Algorithm 1?

Typos:
1. Line 173: $\mathbb{E}\left[f\left(U^{N}\right) \mid X^{1: N}, I^{N}\right]=J_{\varpi, N}^{\mathrm{NEO}}(f)$ to $
\mathbb{E}\left[f\left(U^{N}\right)\right]=J_{\varpi, N}^{\mathrm{NEO}}(f)
$
2. Line 188: $
\mathbb{P}\left(Y_{n} \in \mathrm{A} \mid Y_{n-1}\right)=P\left(Y_{n}, \mathcal{A}\right)
$ to $
\mathbb{P}\left(Y_{n} \in \mathrm{A} \mid Y_{n-1}\right)=P\left(Y_{n-1}, \mathcal{A}\right)
$

Update after discussion: I thank the authors for answering my questions and I decided to keep my score.

**Time Spent Reviewing:**

1 day

---

> ### Author Response · Authors · 2021-08-10
> **Official comment to reviewer ckCN**
>
> Indeed, there is a typo and we are grateful to you for noticing it.  The Markov kernel should be written for $i \in [2:N]$
> $$r_i(x^i,x^{1:N \setminus \{i\}}) =
>  \prod_{j=1}^{i-1} m(x^{j+1},x^{j})\prod_{j=i+1}^N m(x^{j-1},x^j)$$
>  and for $i = 1$:
>  $$ r_1(x^1, x^{1:N \setminus \{1\}}) = \prod_{j = 2}^N m(x^{j-1}, x^j).$$
>  As for the condition in line $226$, note that for any $i \in [N]$, we have by using repeatedly the fact that $m$ is reversible with respect to $\rho$
> $$     \rho(x^i) \prod_{j = i+1}^N m(x^{j-1}, x^j) = \prod_{j = i+1}^N m(x^j, x^{j-1}) \rho(x^N).$$
> Therefore
> $$ \rho(x^i) r_i(x^i,x^{1:N \setminus \{i\}}) = \prod_{j = 1}^{N-1} m(x^{j+1}, x^j) \rho(x^N)$$
> and since the right hand side does not depend on $i$, we get the result.
>
> In Section 2, we derived non-asymptotic bounds (in particular Theorem 1 and Theorem 2) on the estimators obtained using NEO. These results can be understood as efficiency guarantees as they directly link estimation errors with $N$ and $E_T^\varpi$, which can be used to tune parameters empirically. We did not use any information about the transform $\mathrm{T}$ to get tighter bounds as this is still an area of ongoing research.

---

> > ### Comment · Reviewer_ckCN · 2021-08-31
> > **Response to Authors**
> >
> > I thank the authors for the reply. However, my second question was not entirely conveyed to the authors. Theorem 1 gives bounds for NEO-IS with $f=L$, Theorem 1 gives bounds for NEO-SNIS estimator.
> > Is there any theoretical guarantee for NEO-IS with $f$ other than $L$?
> > Can this result be derived from Theorem 2 with $L=1$?

---

> > > ### Author Response · Authors · 2021-09-01
> > > **Response to reviewer  ckCN**
> > >
> > > The function $\mathrm{L}$ is generic in Theorem 1. Even though the main case of interest is usually the computation of the normalizing constant which motivated our presentation of Theorem 1. Thus, the guarantees provided by Theorem 1 could be obtained for any $f$. Similarly, as you underline it, these guarantees could be obtained by enforcing $L = 1$, which then imply $Z = 1$ and $\pi(f) = \int f(x)\rho(dx)$ and then using Theorem 2.

---

### Official Review · Reviewer_dfab · 2021-07-17

**Rating:** 7
**Confidence:** 2

**Summary:**

This paper studies the problem of sampling from a complicate distribution and approximating the partition function. The authors derive a novel family of importance samplers and MCMC samplers.

**Limitations And Societal Impact:**

As a minor suggestion, it would be nice to have more discussions on the differences and improvements of NEO-IS over NEIS in the introduction section.

**Main Review:**

This paper studies the problem of generating samples from a complicated distribution of the form pi(x) promotional to p(x)L(x) and computing the normalizing constant of this distribution. For example, in Bayesian inference p is the prior distribution, pi is the posterior distribution, and L corresponds to the likelihood function. Previous approaches for this task are based on importance sampling, including annealed importance sampling, sequential Monte Carlo, and neural importance sampling. More recently, [23] proposed the non-equilibrium importance sampling (NEIS) which transporting the samples from the reference distribution p using an homogeneous differential flow, and the estimator of the normalizing constant is the integral over the whole orbit instead of only the endpoints.

The authors propose an improved algorithm (NEO-IS) than NEIS. The samples from the reference distribution are propagated under a discrete-time dynamical system, and the estimator of the normalizing constant is obtained as a reweighed integral over the whole orbit using the importance sampling rule. This estimator is unbiased in contrast to NEIS. Building upon NEO-IS for computing the normalizing constant, the authors also propose NEO-MCMC for approximate sampling. Both theoretical guarantees and numerical experiments are provided to show the outstanding performance of these methods.

**Time Spent Reviewing:**

1

---

> ### Author Response · Authors · 2021-08-10
> **Official comment to Reviewer dfab**
>
> Thank you for this review and positive feedback.  Our algorithm is indeed inspired by NEIS. Section 4 details how NEO-IS can be adapted to continuous-time dynamical systems and contains a thorough description of NEIS supported by additional results in Appendix B. This will be stated more clearly in the introduction in the final version to avoid any confusion.

---

### Official Review · Reviewer_JCRj · 2021-07-17

**Rating:** 6
**Confidence:** 2

**Summary:**

This paper proposes an algorithm that estimates the normalizing constant of a distribution of the form $\pi(x)\propto\rho(x)L(x)$ given samples from $\rho(x)$. It further extends the algorithm to a sampling algorithm on the distribution.

**Limitations And Societal Impact:**

Sec 6 discusses some potential future work, but there is not much discussion on limitations. There is no foreseeable potential negative societal impact.

**Main Review:**

The normalizing constant estimating algorithm NEO-IS is based on the idea of NEIS, introduced in a previous work. NEIS can suffer from bias when the discretization step is not small enough. NEO-IS reduces the bias of NEIS by using a proposal density in a discrete sum form. While the discrete proposal density reduces the bias, does it make it harder to find a $\rho_T$ that approximates $\pi$? Does it increase $\mathbb{E}_T^{\varpi}$?

Another contribution of the paper is a new sampling algorithm, NEO-MCMC. The authors of NEIS did not extend it to a sampling algorithm.

NEO-IS and NEO-MCMC are evaluated experimentally in sec 5. For NEO-IS, experiments on NEO vs. NEIS are only performed on low dimensional MG25 and are only presented in the appendix, while experiments on NEO vs. other algorithms are performed on higher dimensional MG25 and a few other distributions. How does NEO-IS compare with NEIS on sampling from other distributions?

This paper, especially the mathematical part of it, is not easy to follow. It would be much easier for the readers if when introducing new math equations, the authors could discuss more about their motivation and intuition.

Some typo: In line 85, should it be Appendix A.2?

Overall, I think this paper brings some interesting new ideas, but it needs more work on presentation and comparison with previous works.


**Time Spent Reviewing:**

5

---

> ### Author Response · Authors · 2021-08-10
> **Official comment to Reviewer JCRj**
>
> Thanks for your review. Contrary to NEIS, NEO-IS actually always returns an unbiased estimate of the normalizing constant as established in Theorem 1. We leverage the unbiasedness of the NEO-IS normalizing constant estimate to further obtain NEO-MCMC, a novel MCMC algorithm to sample from a given target. As you rightly pointed out, the quantity $E_T^\varpi$ indeed plays a key role in the analysis of NEO-IS as highlighted by Theorem 1 and Theorem 2. Therefore, a good practice to tune NEO-IS (i.e. to tune $\rho_T$) is to find parameters that minimize $E_T^\varpi$.  On the contrary, the bias of the NEIS estimate heavily depends on the discretization of (16) as well as on the selected stopping times, which need to be tuned manually. Hence, for a similar transform, we found in the considered examples that it is much easier to tune the parameters of NEO compared to those of NEIS.
>
> In Section 3, we show how our NEO-IS framework can be adapted to continuous-time dynamical systems. Equations (16) and (17) provide a NEO estimator in this setting which requires a discretization step to obtain a tractable estimator. The discretized version is unbiased for any step-size $h$ and converges to the continuous version as $h$ converges to 0. This is a crucial difference with NEIS whose discretization step (in the state-variable $x$) leads to a biased estimator.
>
> Regarding the experiment related to NEO vs NEIS, while the dimensions are much lower (there is a typo in the supplementary, the dimensions are respectively 5 and 10) than what is provided in the main paper, the setting is much more difficult as the variance is the same for all the dimensions and is equal to 0.005. Indeed, we could have performed the comparisons on other distributions but in order to be fair, we did the comparison only on MG25 because we used the code provided by the authors of the NEIS paper which is specifically tuned for Gaussian mixtures. It would indeed be relevant to also compare NEO with NEIS in higher dimensional settings as done in the main paper. We will add these experiments in the updated version of this paper.
>
> We hope that the additional discussions suggested by all reviewers will improve readability and highlight the novelty of our contributions.

---

> > ### Comment · Reviewer_JCRj · 2021-08-23
> > **Response to authors**
> >
> > Thank you for your response. I look forward to seeing the complete experiments in the updated version. Updated score: 5-> 6.

---

### Official Review · Reviewer_2Bce · 2021-07-21

**Rating:** 7
**Confidence:** 4

**Summary:**

The paper consider IS estimator along the path of the dynamical system

**Main Review:**

The idea is natural and technically sounds. However, still I don't understand what is difference with [23], as only given example is conformal Hamiltonian. Possibly, I miss one, will be happy to adjust score in this case. Other concern is relation to the Orbital MCMC https://arxiv.org/abs/2010.08047 Specially, Section 3.3 of it gives the recipes to construct discrete functions with orbits, which I believe is crucial drawback of this work.

----
Upd. 6->7.

**Time Spent Reviewing:**

4

---

> ### Author Response · Authors · 2021-08-10
> **Official comment to Reviewer 2Bce**
>
> NEIS is a continuous-time algorithm focusing on normalizing constant estimation. It is true that NEO-IS extends the NEIS introduced in [23]. However, contrary to NEO-IS, it is unclear how one could obtain an unbiased estimate of the normalizing constant from NEIS and the numerical implementations provided in [23] return biased estimates. Additionally, [23] does not propose an MCMC algorithm which converges towards a given target. We do propose in this paper an original NEO-MCMC sampler which builds upon the unbiasedness of the NEO-IS estimate.
>
> Thanks for pointing out this interesting work. Orbital MCMC is different in spirit from our work as it relies on a dimensionality augmentation scheme to produce new proposal moves (on the orbit of a deterministic transform indeed!) from the current point, while our approach is closer in principle to an ISIR scheme. As suggested, we will add a discussion and comparison with respect to Orbital MCMC in the revised version of the paper.

---

> > ### Comment · Reviewer_2Bce · 2021-08-23
> > **Re-grade**
> >
> > Thanks for your answer. I re-read the paper and found your claims correct. I should admit my understanding wasn't fully correct. Hence, I adjust my score. However, I would like to say that representation (specially in Sec.2) of the idea should be improved: some technical details could be skipped and that space then can be filled by providing general intuition.

---

### Author Response · Authors · 2021-09-01
**Update on revised version of our paper**

Dear reviewers,

Following your recommendations, we have updated the presentation of the paper and  added comparisons to some new methods: Stochastic Normalizing Flows [1], Orbital MCMC [2] and Contour SGLD [3]. We added SNFs [1] to the importance sampling benchmarks and [1,2,3] to the sampling tasks. In addition, we also performed the benchmarks on the banana-shaped distribution. Since we cannot add figures in the comment section, we only provide the importance sampling benchmark and leave the MCMC figures for the updated version of our paper.

We would like to add some comments regarding the competitors.

- While [1] is a clear competitor for sampling, their importance sampling estimators suffer from high variance. This is due to the fact that they perform importance sampling in the path space because their importance density $p_X$ is intractable. It is well known [4] that the performance of importance sampling heavily depends on the KL divergence $D_{KL}(\mu_X(x) \parallel p_X(x))$ between the target density $\mu_X$ and the importance density $p_X$. In this paper, they use $\mu_X(x)P_f(z \to x)$ as target density and $\mu_Z(z)P_b(z \to x)$ as importance density, where $z$ is a sample from the prior distribution, $x$ is the last sample of the forward path and $P_f$ and $P_b$ are the conditional forward / backward path probabilities, see (8) in [1]. The performance of their IS estimator depends on $D_{KL}(\mu_X(x)P_f(x \to z) \parallel \mu_Z(z)P_b(z \to x))$ which is an upper bound of $D_{KL}(\mu_X(x) \parallel p_X(x))$ (as noted in (15) in their paper). This yields an IS estimator with poor performance. This is confirmed by the following normalizing constant estimation experiment. We consider the MG25 given in our paper and the banana-shaped distribution with log density proportional to $ \sum_{i = 1}^{\lfloor d/2 \rfloor} \big[ - (y_{2i -1} + y_{2i})^2 - y_{2i}^2 \big] / 3 $. We estimate the normalizing constants that are set to 10. The vanilla IS estimator is also provided. The numbers are given in log scale and in (Median, Interquartile range) format (hence the target value is $\log(10) \approx 2.30$).
The SNF uses 3 RealNVP blocks with three hidden layers of dimension 64 and  20 Metropolis Hastings steps per block using a Gaussian proposal density with standard deviation 0.5 (larger standard deviation have been tried but we didn't observe a significant increase in performance). The training is done using a combination of both KL divergences. Details about how this is done is given in (9) in the appendix of [1]. The implementation as well as the training hyperparameters are provided in the public repository of the authors (\url{https://github.com/noegroup/stochastic_normalizing_flows}).
As for NEO, we use the same parameters for MG25 as in the main paper and for the banana-shaped distribution we use the following parameters: $K = 20, \gamma = 0.01, h = 0.3$ and $M = \mathrm{I}_d$. The importance sampling estimates are computed using 50000 samples for both methods, which have approximately the same computational cost for this task.

- GM25:
| dim | NEO         | SNF          | IS            |
|-----|-------------|--------------|---------------|
| 10  | 2.23 , 0.20 | -8.97, 0.51  | -10.07, 1.50  |
| 20  | 2.08, 0.21  | -9.30, 0.48  | -12.34, 0.43  |
| 45  | 1.91, 0.23  | -11.47, 0.60 | - 14.07, 0.41 |

- Banana-shaped:
| dim | NEO         | SNF          | IS            |
|-----|-------------|--------------|---------------|
| 10  | 2.17, 0.10  | -11.22, 0.21 | 2.01, 0.14    |
| 15  | 2.05, 0.15  | -19.87, 0.13 | 1.86, 0.20    |
| 20  | 1.81, 0.19  | -20.13, 0.17 | 1.57, 0.32    |

As to the sampling benchmarks, [1,2,3] were not able to recover all of the modes on dimensions larger than $10$ on GM25. Overall, [1] performs better than [2,3] on higher dimensions: it explores $\approx 10$
modes (out of  25) in dimension 10 while [2,3] recover less than 5 on average.
On the other hand, [1] performs slightly better than NEO on the funnel distribution considered in our experiment section but with a greater computational cost due to the optimization of the Real-NVP blocks.
We have used the publicly available code for [2,3]: \url{https://github.com/WayneDW/Contour-Stochastic-Gradient-Langevin-Dynamics}, \url{https://github.com/necludov/oMCMC}.

[1] Stochastic Normalizing Flows. Hao Wu, Jonas Köhler, Frank Noé

[2] Orbital MCMC. Kirill Neklyudov, Max Welling

[3] A Contour Stochastic Gradient Langevin Dynamics Algorithm for Simulations of Multi-modal Distributions. Wei Deng, Guang Lin, Faming Liang

[4] The sample size required in importance sampling. Sourav Chatterjee, Persi Diaconis

---

### Decision · Program_Chairs · 2021-09-28

**Decision:**

Accept (Poster)

**Comment:**

The paper develops a new, normalizing constant estimation method which is unbiased, and leverages this to achieve improved sampling methods for the same family of distributions. From a technical perspective the insights are definitely above the bar. The main concern that the reviewers and I shared is the presentation of the paper and the lack of intuition provided for some of the more technical aspects of it. While I understand that space is limited, I encourage the authors to expand upon the exposition in Section 2 in particular. Additionally, the experimental evaluation was lacking, however, in the discussions, the authors provided additional experiments and context which I think will improve the paper substantially.

**Consistency Experiment:**

NeurIPS has a long history of experimentation. In 2014, NeurIPS ran an experiment in which 10% of submissions were reviewed by two independent committees to quantify the randomness in the review process. This year, we repeated a variant of this experiment to see how the quality of the review process has changed over time.  This paper was part of the experiment and was therefore assigned to two committees (consisting of reviewers, an Area Chair, and a Senior Area Chair) that reached independent decisions.  If both committees made the same recommendation, this recommendation was followed. If a single committee recommended acceptance, the paper was accepted (with the exception of a few cases in which the other committee identified what we considered a fatal flaw, e.g., an error in a key result).

Both committees reached the same decision: **Accept (Poster)**

The other committee assigned to the paper recommended **Accept (Poster)**.  You can find the other set of reviews, along with any follow up discussion with the authors here:
https://openreview.net/forum?id=PTo9C5G0qK9